

# ChimpACT: A Longitudinal Dataset for Understanding Chimpanzee Behaviors

**Xiaoxuan Ma** [1,★]
maxiaoxuan@pku.edu.cn

**Stephan P. Kaufhold** [2,★]
spkaufho@ucsd.edu

**Jiajun Su** [1,★]
sujiajun@pku.edu.cn

**Wentao Zhu** [1]
wtzhu@pku.edu.cn

**Jack Terwilliger** [2]
jterwilliger@ucsd.edu

**Andres Meza** [2]
anmeza@ucsd.edu

**Yixin Zhu** [3,5,✉]
yixin.zhu@pku.edu.cn

**Federico Rossano** [2,✉]
frossano@ucsd.edu

**Yizhou Wang** [1,3,4]
yizhou.wang@pku.edu.cn

★ X. Ma, S. Kaufhold, and J. Su contributed equally.     ✉ Corresponding authors
[1] CFCS, School of Computer Science, Peking University, China
[2] Department of Cognitive Science, University of California, San Diego, USA
[3] Institute for Artificial Intelligence, Peking University, China
[4] Nat'l Eng. Research Center of Visual Technology, China
[5] PKU-WUHAN Institute for Artificial Intelligence, China

https://shirleymaxx.github.io/ChimpACT/

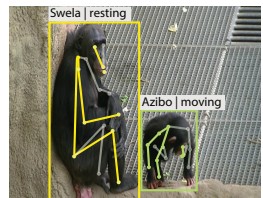 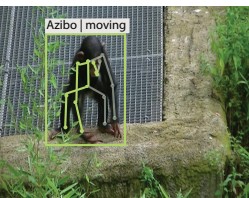 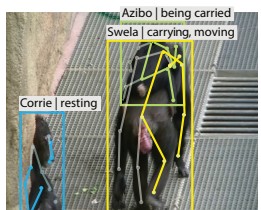 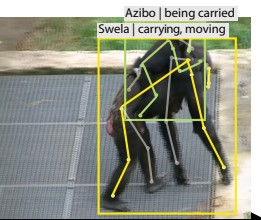

Figure 1: **Sample frames and annotations from a ChimpACT clip.** While we also annotate visibility for both the bounding box and the keypoint, these are omitted here for clarity.

## Abstract

Understanding the behavior of non-human primates is crucial for improving animal welfare, modeling social behavior, and gaining insights into distinctively human and phylogenetically shared behaviors. However, the lack of datasets on non-human primate behavior hinders in-depth exploration of primate social interactions, posing challenges to research on our closest living relatives. To address these limitations, we present ChimpACT, a comprehensive dataset for quantifying the longitudinal behavior and social relations of chimpanzees within a social group. Spanning from 2015 to 2018, ChimpACT features videos of a group of over 20 chimpanzees residing at the Leipzig Zoo, Germany, with a particular focus on documenting the developmental trajectory of one young male, Azibo. ChimpACT is both comprehensive and challenging, consisting of 163 videos with a cumulative 160,500 frames, each richly annotated with detection, identification, pose estimation, and fine-grained spatiotemporal behavior labels. We benchmark representative methods of three tracks on ChimpACT: (i) tracking and identification, (ii) pose estimation, and (iii) spatiotemporal action detection of the chimpanzees. Our experiments reveal that ChimpACT offers ample opportunities for both devising new methods and adapting existing ones to solve fundamental computer vision tasks applied to chimpanzee groups, such as detection, pose estimation, and behavior analy-

37th Conference on Neural Information Processing Systems (NeurIPS 2023) Track on Datasets and Benchmarks.

sis, ultimately deepening our comprehension of communication and sociality in non-human primates.

# 1 Introduction

Studying the behavior of non-human primates is essential for gaining evolutionary insights (Langer-graber et al., 2012), conducting biomedical research (Schapiro et al., 2005), and improving animal welfare (Dawkins, 2003; Gonyou, 1994). Furthermore, given the close phylogenetic proximity between humans and non-human primates, it provides an ethically sound and effective avenue to probe the roots of human sociality (The Chimpanzee Sequencing and Analysis Consortium, 2005). Traditional field research typically requires researchers to enter wildlife conservation areas for extended durations, sometimes spanning multiple years. This involves habituating primate groups to human presence, capturing video footage, and laboriously manually coding these videos for subsequent statistical analysis (Hobaiter et al., 2017; Fröhlich et al., 2020; Surbeck et al., 2017; Luncz et al., 2018; Sirianni et al., 2015). While video coding is heralded as the gold standard for distilling rich, nuanced behavioral patterns (Wiltshire et al., 2023), its practical utility hinges on the efficiency of the coding process. This not only demands researchers with specialized expertise but is also prone to attentional biases.

Recent strides in computer vision offer promise for the automated analyses of non-human primate behaviors, especially those of chimpanzees. Nevertheless, the scarcity of high-quality longitudinal datasets remains a bottleneck. Assembling chimpanzee behavioral data is a formidable endeavor, necessitating substantial resources and expertise. This process entails continuous video recording and meticulous manual annotation, with a keen emphasis on annotation accuracy and consistency. While some datasets (Marks et al., 2022; Bala et al., 2020) confine subjects to indoor enclosures, resulting in atypical and constrained environments, others resort to sourcing and labeling chimpanzee images online (Labuguen et al., 2021; Desai et al., 2022; Ng et al., 2022; Yao et al., 2023). Unfortunately, these often overlook the intricate social dynamics inherent to chimpanzee groups, hindering a comprehensive study of their social behaviors and social relationships.

Addressing the existing limitations, we introduce `ChimpACT`, a comprehensive longitudinal dataset tailored for the in-depth study of chimpanzee social behavior in a semi-naturalistic setting, replete with annotations of instance bounding boxes, body poses, and spatial-temporal action labels. A comparison with other datasets is provided in Tab. 1. `ChimpACT` encompasses footage of a specific chimpanzee group residing at Leipzig Zoo, Germany, with a particular focus on a juvenile male named Azibo (refer to Fig. 1). The data, gathered between 2015 and 2018, employs *focal sampling*

Table 1: **Comparison of `ChimpACT` with existing primate behavioral datasets.** Square-bracketed numbers denote label counts for the chimpanzee category. ⊘ denotes undocumented. For the "Species" row, G represents general, P for primates, M for macaque, and C for chimpanzee. In the "Source" row, I stands for Internet, Z for zoo, C for cage, W for wild, and CP for captive.

| Dataset | Species | Track 1 detection, tracking, ReID | | | | Track 2 pose estimation | | | | Track 3 action recognition | | Source |
| | | ID # | frame # | box # | track | frame # | pose # | track | dim. | class # | label # | |
|---|---|---|---|---|---|---|---|---|---|---|---|---|
| AP-10K (Yu et al., 2021) | G | ✗ | ✗ | ✗ | ✗ | 10,015 | 13,028 [<500] | ✗ | 2D | ✗ | ✗ | I |
| AnimalKingdom (Ng et al., 2022) | G | ✗ | ✗ | ✗ | ✗ | 33,099 | 99,297 [576] | ✗ | 2D | 140 | 30,100 [⊘] | I |
| OpenApePose (Desai et al., 2022) | P | ✗ | ✗ | ✗ | ✗ | 71,868 | 71,868 [18,010] | ✗ | 2D | ✗ | ✗ | I |
| OpenMonkeyChallenge (Yao et al., 2023) | P | ✗ | ✗ | ✗ | ✗ | 111,529 | 111,529 [<10,000] | ✗ | 2D | ✗ | ✗ | I & Z |
| OpenMonkeyStudio (Bala et al., 2020) | M | ✗ | ✗ | ✗ | ✗ | 194,518 | 33,192 [0] | ✓ | 3D | ✗ | ✗ | C ($6.7m^2$) |
| MacaquePose (Labuguen et al., 2021) | M | ✗ | ✗ | ✗ | ✗ | 13,083 | 16,393 [0] | ✗ | 2D | ✗ | ✗ | I & Z |
| SIPEC (Marks et al., 2022) | M | 4 | 191 | 2,200 [0] | ✓ | ✗ | ✗ | ✗ | ✗ | 4 | ⊘ | C ($15m^2$) |
| CCR (Bain et al., 2019) | C | 13 | 936,914 | 1,937,585 | ✓ | ✗ | ✗ | ✗ | ✗ | ✗ | ✗ | W |
| **`ChimpACT`** (Ours) | C | 23 | 160,500 | 56,324 | ✓ | 16,028 | 56,324 | ✓ | 2D | 23 | 64,289 | CP ($4400m^2$) |

(Altmann, 1974). Born in April 2015, Azibo[1] has been living in the group since birth, providing a unique perspective on the development of an individual within a chimpanzee group characterized by well-defined kin relationships. (also depicted in Fig. 2a). The footage covers the daily lives of over 20 chimpanzees in a group, aggregating to 163 video recordings, approximately 160,500 frames, and spanning around 2 hours.

Our annotations on `ChimpACT` are extensive, marking each individual's detection, tracking, identification, pose estimation, and spatiotemporal action detection. Sample frames with their corresponding annotations are illustrated in Fig. 1. Each chimpanzee's identity is confirmed by a seasoned behavioral researcher familiar with the Leipzig chimpanzees, ensuring data precision and trustworthiness. Crucially, we employ an ethogram (detailed in Fig. 2b) devised by the same expert for fine-grained action labels. To our knowledge, `ChimpACT` is the first to furnish ethogram annotations for the machine learning and computer vision community. This bespoke ethogram delineates behaviors into four categories: locomotion, object interaction, social interaction, and others, with each encompassing several detailed actions we diligently annotate.

While advancements in computer vision have notably addressed human-centric tasks, such as human pose estimation (Sun et al., 2019; Xiao et al., 2018), the dearth of chimpanzee datasets has curtailed progress on chimpanzee-specific challenges. Despite their genetic closeness to humans (The Chimpanzee Sequencing and Analysis Consortium, 2005), deciphering chimpanzee behaviors is intricate due to their unique morphology, appearance, and keypoint articulation. Highlighting the importance of crafting sophisticated chimpanzee perception models, we evaluate prominent human perception methods on three tracks: (i) detection, tracking, and identification (ReID), (ii) pose estimation, and (iii) spatiotemporal action detection. Our findings underscore `ChimpACT`'s potential as a platform for the community to pioneer advanced techniques for better perception of the chimpanzees and ultimately contribute to a deeper understanding of non-human primates.

## 2 Related work

**Computer vision for animals**    A myriad of datasets and benchmarks have emerged, harnessing computer vision techniques to advance animal research. For instance, 3D-ZeF20 (Pedersen et al., 2020) introduces 3D tracking of zebrafish to the MOT benchmarks. AnimalTrack (Zhang et al., 2023) emphasizes multi-animal tracking across a spectrum of species. AP-10K (Yu et al., 2021) and APT-36K (Yang et al., 2022) venture into animal pose estimation for diverse species. AnimalKingdom (Ng et al., 2022) extends its focus to fine-grained multi-label action recognition. Moreover, several studies have delved into multi-agent behavior understanding from a social interaction perspective (Sun et al., 2021, 2023). Distinctively, `ChimpACT` stands out as a holistic benchmark, encompassing three varied downstream tasks and boasting rich annotations of social interactions.

**Human video datasets**    In contrast to animal-centric video datasets, a more substantial collection is tailored to human subjects, addressing diverse human-centric video understanding tasks. For instance, the MOT Challenge (Milan et al., 2016) is curated for multi-person tracking. Other benchmarks like COCO (Lin et al., 2014) and MPII (Andriluka et al., 2014) cater to human pose estimation. Meanwhile, datasets such as Kinetics (Kay et al., 2017), ActivityNet (Fabian Caba Heilbron and Niebles, 2015), and AVA (Gu et al., 2018) are dedicated to human action recognition. With `ChimpACT`, we encompass analogous tasks but introduce challenges specific to chimpanzee behavior.

**Datasets on primate behavioral understanding**    Most existing primate datasets are tailored towards individual primate detection and pose estimation. These either stem from confined indoor settings (Bala et al., 2020; Marks et al., 2022) or are amassed and labeled from online sources (Labuguen et al., 2021; Desai et al., 2022; Ng et al., 2022; Yao et al., 2023). The former can induce atypical behavioral patterns, while the latter often omits longitudinal interactions, rendering them suboptimal for analyzing chimpanzee social dynamics. A notable exception is the CCR dataset (Bain et al., 2019), chronicling 13 chimpanzees in the Bossou forest over two years. Yet, it primarily focuses on individual detection and recognition, lacking behavioral annotations, which limits its efficacy for probing the social nuances of wild primates. Tab. 1 offers a comprehensive comparison. The narrow focus of most primate datasets on singular tasks restricts their breadth and adaptability to diverse research inquiries. Contrarily, `ChimpACT` presents a multifaceted approach, encompassing identities, kinship, detection labels, pose annotations, ethograms, and fine-grained action labels. This richness

---

[1]Details about Azibo can be found at `https://tinyurl.com/azibo-chimp/`.

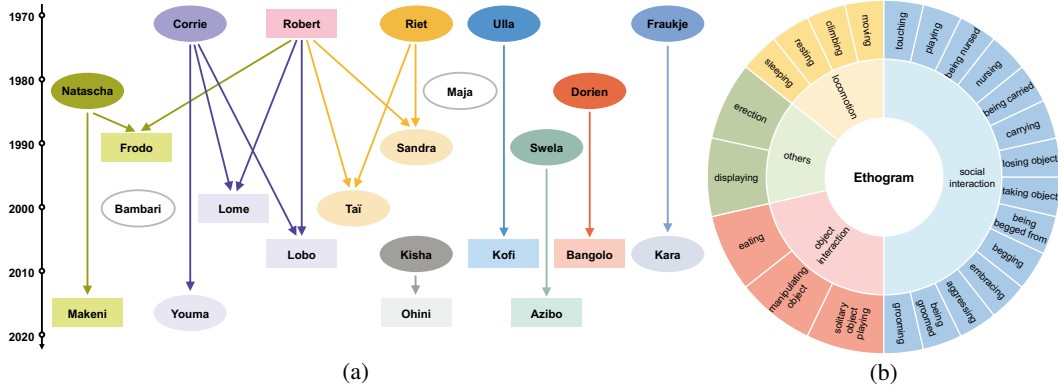

Figure 2: **(a) Kinship of the observed chimpanzee group.** Rectangles and ellipses represent males and females, respectively, with arrows flowing from the parents to the child. Their vertical position relative to the time axis indicates the year of birth. **(b) Ethogram with annotated behaviors.**

positions it as an indispensable tool for devising advanced chimpanzee behavior analysis methods and enriching the overarching comprehension of primate behavior.

**Methods for primate behavioral analysis** Deciphering primate behavior is instrumental in understanding their social dynamics and cognitive abilities. Behavioral analysis often encompasses subtasks like individual detection, tracking, and identification (Bain et al., 2019; Marks et al., 2022), pose estimation (Labuguen et al., 2021; Desai et al., 2022; Mathis et al., 2018; Wiltshire et al., 2023), and behavior recognition (Ng et al., 2022; Bain et al., 2021). While each task has specialized techniques, many are rooted in human behavioral research. Numerous algorithms exist for human tracking (Bewley et al., 2016; Pang et al., 2021), pose estimation (Sun et al., 2019; Xiao et al., 2018), and behavior recognition (Feichtenhofer et al., 2019). However, due to the dearth of primate datasets, primate behavioral analysis often repurposes algorithms designed for humans, including:

- **Detection, tracking, and ReID** identify individual primates in videos, often leveraging established object or human detection algorithms like Mask-RCNN (He et al., 2017). For instance, SIPEC (Marks et al., 2022) employs Mask-RCNN with a ResNet backbone (He et al., 2016) to track and segment macaque. Bain et al. (2019) utilize CNNs to crop and identify individual chimpanzees.

- **Pose estimation** discerns primate poses, frequently adapting human pose estimation methods like SimpleBaseline (Xiao et al., 2018). DeepLabCut (Mathis et al., 2018; Lauer et al., 2022), for instance, employs ResNet-50 with ImageNet pre-trained weights for 2D animal pose estimation. SIPEC (Marks et al., 2022) modifies SimpleBaseline for 2D macaque poses.

- **Behavior recognition** identifies primate actions and interactions. Contemporary methods (Bain et al., 2021; Bohnslav et al., 2021) often derive from human action recognition algorithms like SlowFast (Schindler and Steinhage, 2021). Notably, Bain et al. (2021) integrates audio cues for classifying two simple non-interactive behaviors: nut cracking and buttress drumming. In contrast, `ChimpACT` encompasses over 20 daily behaviors under an ethogram hierarchy, capturing both solitary actions and intricate social interactions.

In essence, primate behavioral analysis is a multifaceted endeavor, intertwining computer vision, machine learning, and primatology. The advent of `ChimpACT` marks a significant stride towards unraveling the intricate social tapestry of our primate kin.

## 3 `ChimpACT`

### 3.1 Dataset description

`ChimpACT` comprises about 2-hour video footage of chimpanzees recorded at the Leipzig Zoo in Germany between 2015 and 2018. The videos focus on one male chimpanzee, Azibo, who was born in April 2015 to Swela and has lived with the A-chimpanzee group[2] at the Leipzig Zoo ever since.

---

[2]The A-chimpanzee group is among the most extensively studied zoo-residing chimpanzee cohorts. Its members have been subjects of both behavioral and cognitive studies, spanning observational and experimental

The longitudinal observation of Azibo offers a rare lens into his behavioral evolution, social dynamics, and intra-group relationships. With over 20 individuals in the group, `ChimpACT` serves as a treasure trove of insights into chimpanzee behavior and social intricacies. Key attributes of `ChimpACT` are delineated below.

**Longitudinal data**   Spanning four years, `ChimpACT` chronicles the life of a stable zoo-residing chimpanzee group, offering a rare glimpse into the nuances of chimpanzee social behavior development. Tracking the growth and interactions of a young chimpanzee within this group sheds light on chimpanzee socialization, the evolution of social skills (Matsuzawa, 2013), the formation of social bonds and integration into the dominance hierarchy (Matsuzawa et al., 2006), and the acquisition of group-specific cultural behaviors (Van Leeuwen, 2021; Musgrave et al., 2021).

**Semi-naturalistic and social environment**   The videos in `ChimpACT` capture chimpanzees in their semi-naturalistic habitats at Leipzig Zoo, split between indoor (96 videos) and outdoor (67 videos) enclosures. The indoor space, spanning roughly 400 $m^2$, features a plethora of environmental enrichments, ranging from wooden climbing structures and hammocks to vegetation and foraging boxes. When weather permits, the chimpanzees have access to a 4000 $m^2$ outdoor area, replete with vegetation, surrounded by an artificial river, and complemented by enrichments similar to the indoor space. This blend of environments ensures the dataset's relevance for both naturalistic and artificial environments. The multifaceted physical and social surroundings of the chimpanzees further imbue the dataset with intricate behaviors and social dynamics.

**Ethogram with solitary and social behaviors**   `ChimpACT` captures the daily life of group-living chimpanzees, offering invaluable insights into the evolution and sustenance of their social behaviors and relationships (Nishida et al., 2010). By focusing on a juvenile chimpanzee, `ChimpACT` illuminates facets of social learning, communication, bonding, and more, all pivotal in the social and ecological life of chimpanzees (Bard et al., 2014). To systematically represent these behaviors, we composed an ethogram—a detailed catalog of behavioral categories, depicted in Fig. 2b (further details in Appx. A). This ethogram organizes behaviors into four primary categories, like locomotion and social interaction, each further subdivided into several fine-grained actions, meticulously annotated and validated with expert oversight. By delving into these behaviors, `ChimpACT` elucidates not only the social dynamics shaping social relationships but also the cognitive and ecological influences on juvenile chimpanzee behaviors.

## 3.2   Dataset collection

The focal video data were collected with the Chimpanzee-A group housed at Leipzig Zoo, Germany, using focal sampling (Altmann, 1974). Videographers were instructed to focus on Azibo and his mother, Swela, but also on capturing the environmental context and his interactions with other chimpanzees. Videos from `ChimpACT` were sampled from a larger set of around 405 hours of longitudinal focal video recordings of the dyad between 2015 and 2018. These videos were recorded by several research assistants during the daytime (7am–4pm) using tripod-mounted RGB cameras. Two JVC Everio camera models were utilized across the years, filming with a framerate of 25 (Codec H.264) and with resolutions of $720 \times 578$ and $1280 \times 720$, respectively. The mother-infant dyad was filmed for about five hours each week during the observation period. The footage contains both optical zoom and camera movements.

## 3.3   Dataset tasks and annotations

`ChimpACT` supports three tracks: (i) chimpanzee detection, tracking, and ReID, (ii) chimpanzee pose estimation, and (iii) spatiotemporal action detection. We provide fine-grained annotations for each track. From the extensive footage, we curated 163 video clips, each approximately 1000 frames in length. Fifteen adept annotators were then tasked with annotating bounding boxes, body keypoints, and fine-grained behavioral classes for each chimpanzee at intervals of every 10 frames. To ensure accuracy and consistency, a behavioral researcher familiar with the chimpanzee group meticulously reviewed and refined the identity and behavioral class annotations. For a deeper dive into the annotation process and its quality, please refer to Appx. A and our dedicated website.

---

designs, conducted by researchers affiliated with the MPI for Evolutionary Anthropology (Baker, 2022; McEwen et al., 2022).

**Detection, tracking, and ReID**  This task encompasses the detection and tracking of individual chimpanzees across video sequences, subsequently coupled with their re-identification. `ChimpACT` features over 23 distinct chimpanzee individuals, each identified by a primate expert familiar with the Leipzig A-group chimpanzees. Initially, annotators were instructed to delineate the bounding box of each chimpanzee, ensuring consistent box IDs for the same individual throughout a video clip. Subsequently, the expert matched these box IDs with the corresponding true names of the chimpanzees, resulting in the identification of 23 unique individuals. Additionally, every annotated bounding box is attached with a visibility attribute, indicating if the chimpanzee is fully visible, truncated, or occluded in a given frame. Such visibility annotations can support the reasoning of the chimpanzee behavior, potentially bolstering tracking robustness. Fig. 3a illustrates the occurrence frequency (on a *log* scale) of each individual, revealing a long-tail distribution. This pattern aligns with the focal sampling strategy, where Azibo is the primary subject. Notably, Swela, Azibo's mother, also exhibits a high occurrence frequency, resonating with prior studies (Boesch, 1996).

**Pose estimation**  Pose estimation aims to predict the locations of the chimpanzee joints that have semantic meaning, such as the knee and shoulder, from an input image. There are four keypoints on the chimpanzee's face (*i.e.*, two for the eyes, and one each for the upper and lower lips), for a total of 16 chimpanzee keypoints (refer to Sec. 3.3 and Fig. 4). Annotators are tasked with marking the 2D joint coordinates and the visibility status of each joint. We adopt the visibility protocol from the COCO 2D human keypoint annotations (Lin et al., 2014), where a value of 0 indicates a joint outside the image frame, 1 signifies an obscured joint within the image, and 2 designates a clearly visible joint. Such an annotation protocol affords rea-

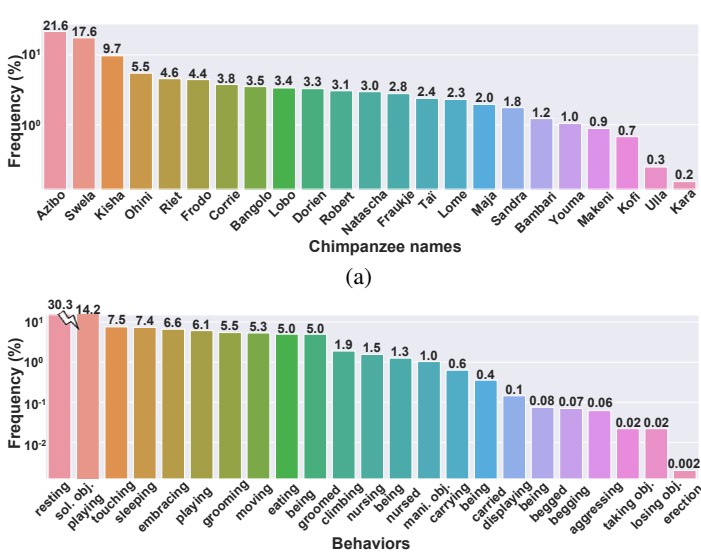

(a)

(b)

Figure 3: **(a) Distribution (in log scale) of annotations for each individual. (b) Distribution (in log scale) of annotations for each behavior.** Vector graphics; zoom for details.

son about chimpanzee's orientation and action based on facial joint visibility. For instance, the chimpanzee might be eating something if the two lips are apart. Sample frames showcasing pose annotations are depicted in Fig. 1. Notably, `ChimpACT` holds the potential for future expansion to encompass pose tracking tasks, analogous to the PoseTrack (Andriluka et al., 2018) for humans.

Table 2: **Keypoint definitions for chimpanzee.**

| No. | Definition | No. | Definition |
|---|---|---|---|
| 0 | Root of hip | 8 | Right eye |
| 1 | Right knee | 9 | Left eye |
| 2 | Right ankle | 10 | Right shoulder |
| 3 | Left knee | 11 | Right elbow |
| 4 | Left ankle | 12 | Right wrist |
| 5 | Neck | 13 | Left shoulder |
| 6 | Upper lip | 14 | Left elbow |
| 7 | Lower lip | 15 | Left wrist |

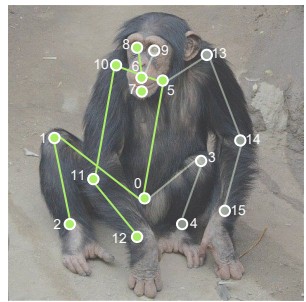

Figure 4: **Keypoint definitions for chimpanzee.**

**Spatiotemporal action detection**  Spatiotemporal action detection seeks to attribute one or multiple behavioral labels to each bounding box containing a chimpanzee, leveraging the spatiotemporal context within a video clip. Our ethogram, detailed in Fig. 2b, delineates 23 nuanced subcategories of behaviors and guides the fine-grained annotations of chimpanzee behavior, such as "climbing"

within the "locomotion" category. Notably, within the realm of social interactions, we meticulously differentiate between the action performer and receiver. For instance, the grooming behavior is bifurcated into "grooming" and "being groomed." Every chimpanzee in a frame has its subcategory behavior annotated. It is not uncommon for an individual to simultaneously exhibit multiple behaviors, exemplified by Swela's "carrying" and "moving" actions in Fig. 1. The distribution of these behavioral annotations, visualized in Fig. 3b on a *log* scale, reveals a long-tail distribution, mirroring the authentic behavioral tendencies of chimpanzees in their natural habitats.

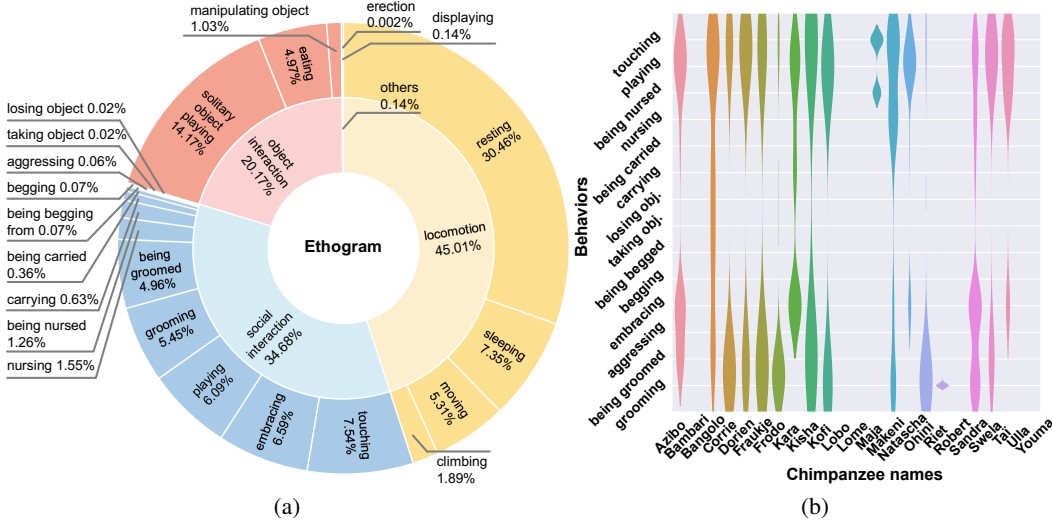

Figure 5: **(a) Distribution of the annotated behavior categories. (b) Distribution showcasing individuals alongside their respective social behaviors.** Vector graphics; zoom for details.

Fig. 5a showcases the distribution of the annotated behaviors, with social interactions constituting approximately 35% of the total annotations. Furthermore, Fig. 5b delineates the distribution of labeled social behaviors across distinct individuals, highlighting grooming, playing, and touching as predominant activities within the social dynamics of the group-living chimpanzees.

In essence, `ChimpACT` emerges as an invaluable resource for researchers spanning the domains of primatology, comparative psychology, computer vision, and machine learning. It furnishes a comprehensive and varied array of annotations, paving the way for in-depth analysis of multifaceted chimpanzee behaviors and catalyzing the development of advanced machine learning algorithms. The inherent long-tail distribution not only presents a formidable challenge for chimpanzee identification and behavior recognition but also beckons explorations into few-shot learning in future endeavors.

## 4 Experiments

To rigorously assess `ChimpACT`, we benchmark a suite of representative methods across the aforementioned three tracks: (i) detection, tracking, and ReID, (ii) pose estimation, and (iii) spatiotemporal action detection. Our computational framework leverages four NVIDIA GeForce RTX 3090 GPUs (24GB) for both training and evaluation across all tracks. In the subsequent sections, we delve into the implementation details, baseline methods, and evaluation metrics for each track.

### 4.1 Detection, tracking, and ReID

**Setting** We evaluate several prominent Multiple Object Tracking (MOT) algorithms on `ChimpACT`, including both classical methods such as SORT (Bewley et al., 2016), DeepSORT (Wojke et al., 2017), and Tracktor (Bergmann et al., 2019), as well as the state-of-the-art methods such as ByteTrack (Zhang et al., 2022), and OC-SORT (Cao et al., 2023). All implementations are based on the MMTracking (Contributors, 2020a) codebase. For those methods supporting flexible detection backbones, we trial two typical detectors, including the two-stage detector Faster R-CNN (Ren et al., 2015) and the one-stage detector YOLOX (Ge et al., 2021). Each method undergoes training for 10 epochs, adhering to the official configurations, which encompass optimizer settings, batch size, data augmentation techniques, and pre-trained models. Given that the three classical methods lack inherent ReID

Table 3: **Results of the detection, tracking, and ReID track on the `ChimpACT` test set.** The row highlighted in light blue is the performance reference on the human tracking dataset MOT-17 (Milan et al., 2016). − denotes not applicable. ⊘ denotes unreported.

| Method | Detector | ReID | HOTA ↑ | MOTA ↑ | MOTP ↑ | IDF1 ↑ | mAP ↑ | nFP ↓ | nFN ↓ | nIDs ↓ |
|---|---|---|---|---|---|---|---|---|---|---|
| SORT (Bewley et al., 2016) | Faster R-CNN | ResNet-50 | $42.6_{+1.0}$ | $47.4_{+0.6}$ | $22.9_{+1.3}$ | $42.7_{+1.2}$ | $70.7_{+1.6}$ | $19.1_{+0.3}$ | $31.4_{+0.5}$ | $2.1_{+0.0}$ |
| | YOLOX | | $39.8_{+0.8}$ | $43.2_{+1.3}$ | $20.3_{+0.5}$ | $37.7_{+1.7}$ | $71.4_{+1.6}$ | $16.1_{+3.1}$ | $37.8_{+1.7}$ | $2.8_{+0.5}$ |
| DeepSORT (Wojke et al., 2017) | Faster R-CNN | ResNet-50 | $47.6_{+0.4}$ | $46.7_{+0.5}$ | $23.0_{+1.2}$ | $52.8_{+1.5}$ | $70.7_{+1.6}$ | $19.0_{+0.3}$ | $31.4_{+0.5}$ | $2.9_{+0.1}$ |
| | YOLOX | | $40.2_{+1.0}$ | $43.2_{+1.2}$ | $20.3_{+0.5}$ | $38.4_{+1.9}$ | $71.4_{+1.6}$ | $16.1_{+3.1}$ | $37.8_{+1.7}$ | $2.9_{+0.6}$ |
| Tractor (Bergmann et al., 2019) | Faster R-CNN | ResNet-50 | $49.5_{+0.7}$ | $50.5_{+1.1}$ | $22.6_{+1.1}$ | $55.6_{+1.2}$ | $70.7_{+1.6}$ | $13.8_{+0.5}$ | $35.2_{+0.7}$ | $0.5_{+0.0}$ |
| QDTrack (Pang et al., 2021) | Faster R-CNN | − | $50.3_{+3.2}$ | $54.2_{+4.6}$ | $22.2_{+1.4}$ | $55.8_{+3.6}$ | $77.8_{+2.0}$ | $19.7_{+3.6}$ | $24.6_{+0.8}$ | $1.4_{+0.2}$ |
| ByteTrack (Zhang et al., 2022) | Faster R-CNN | − | $43.7_{+0.3}$ | $36.9_{+2.2}$ | $24.6_{+0.3}$ | $48.8_{+1.3}$ | $68.2_{+1.1}$ | $27.7_{+1.1}$ | $34.2_{+1.0}$ | $1.2_{+0.2}$ |
| | YOLOX | − | $49.2_{+0.8}$ | $43.9_{+1.3}$ | $20.3_{+1.0}$ | $55.2_{+1.1}$ | $70.3_{+1.0}$ | $18.0_{+7.4}$ | $37.4_{+6.1}$ | $0.7_{+0.0}$ |
| OC-SORT (Cao et al., 2023) | Faster R-CNN | − | $43.4_{+1.0}$ | $38.2_{+1.9}$ | $24.3_{+0.2}$ | $48.7_{+2.2}$ | $68.7_{+0.8}$ | $25.0_{+1.6}$ | $35.6_{+1.5}$ | $1.2_{+0.1}$ |
| | YOLOX | − | $47.9_{+0.4}$ | $42.1_{+2.6}$ | $20.5_{+0.8}$ | $53.3_{+0.8}$ | $70.5_{+0.8}$ | $20.3_{+1.3}$ | $36.6_{+2.1}$ | $1.1_{+0.3}$ |
| OC-SORT (Cao et al., 2023) | YOLOX | − | 63.2 | 78.0 | ⊘ | 77.5 | ⊘ | 2.7 | 19.0 | 0.3 |

modules, we supplement with a dedicated ReID network built on ResNet-50 (He et al., 2016). The training curves of select methods (refer to Fig. A2a) affirm convergence within the training epochs.

We split the video clips in `ChimpACT` into 80% train, 10% validation, and 10% test. Both the train set and test set cover all the individuals. Models are trained on the training set, with performance metrics reported on the test set. We employ widely-accepted evaluation metrics, drawing from convention in human/object detection, tracking, and ReID (Bewley et al., 2016; Pang et al., 2021; Zhang et al., 2022). Specifically, we utilize (i) mean Average Precision (mAP) Lin et al. (2014) to gauge the detection accuracy, and (ii) the CLEAR metrics (Bernardin and Stiefelhagen, 2008) (MOTA, MOTP, FP, FN, IDs), IDF1 (Ristani et al., 2016), and HOTA (Luiten et al., 2021) to evaluate various facets of the tracking performance. It is worth noting that for FP, FN, and IDs, we report normalized values and denote these metrics as nFP, nFN, and nIDs, respectively.

**Results** Tab. 3 summarizes these tracking algorithms' performances on the `ChimpACT` test set. We conducted three runs for each method and reported the average and variance of these metrics. Notably, the variance across multiple runs is minimal, underscoring the robust reproducibility of our benchmarking. A holistic view of the results reveals that QDTrack (Pang et al., 2021) emerges as the top performer. However, it does suffer from a higher count of identity switches compared to other methods. In terms of detection performance, the YOLOX algorithm (Ge et al., 2021) stands toe-to-toe with Faster R-CNN (Ren et al., 2015). A discernible trend is evident among contemporary tracking methods, which seem to excel in identity association capabilities over their classical counterparts. This is corroborated by marked improvements in tracking metrics like IDF1 and IDs. Such a trend intimates that the latest tracking methods might be adept at maintaining consistent object identities, a pivotal aspect when tracking and analyzing individual trajectories within chimpanzee cohorts.

While the results garnered by the array of tracking algorithms are commendable, they still lag behind the benchmarks set on human-centric datasets (Zhang et al., 2022; Pang et al., 2021; Cao et al., 2023). This disparity can be attributed to challenges like the low contrast and low color variation of the body fur of chimpanzees, compounded by intricate self-occlusions. Nonetheless, this very observation accentuates the significance of `ChimpACT`. It not only offers a challenging arena for tracking algorithms but also stands as an ideal platform for pioneering and refining tracking methods tailored for chimpanzees and other non-human primates.

## 4.2 Pose estimation

**Setting** We benchmark several state-of-the-art human pose estimation methods on `ChimpACT`, including CPM (Wei et al., 2016), SimpleBaseline (Xiao et al., 2018), HRNet (Sun et al., 2019), DarkPose (Zhang et al., 2020). Broadly, human pose estimation methods can be bifurcated into two primary paradigms: heatmap-based and regression-based. We harness the MMPose (Contributors, 2020b) framework for implementing these methods. Please refer to Appx. C for more implementation details. All the models undergo training for 210 epochs, maintaining the official configurations for optimizers, batch sizes, and learning rates. To gauge any potential model overfitting, we present the validation curve on the AP metric in Fig. A2b, reassuringly suggesting an absence of overfitting.

Table 4: **Results of the pose estimation track on `ChimpACT` test set.** The row highlighted in light blue is the performance reference on the human pose estimation dataset COCO (Lin et al., 2014). ⊘ denotes unreported.

| | Method | Backbone | PCK@0.05 | PCK@0.1 | AP | $AP^{50}$ | $AP^{75}$ | $AP^M$ | $AP^L$ | AR |
|---|---|---|---|---|---|---|---|---|---|---|
| *Regression* | SimpleBaseline (Xiao et al., 2018) | ResNet-50 | $25.3_{+0.5}$ | $46.2_{+0.5}$ | $8.6_{+0.4}$ | $27.4_{+1.3}$ | $3.9_{+0.4}$ | $0.3_{+0.1}$ | $12.5_{+0.5}$ | $17.3_{+0.7}$ |
| | | ResNet-101 | $26.2_{+1.0}$ | $46.4_{+1.1}$ | $8.7_{+0.4}$ | $27.5_{+0.6}$ | $4.2_{+0.5}$ | $0.3_{+0.0}$ | $12.9_{+0.2}$ | $17.7_{+0.4}$ |
| | | ResNet-152 | $26.3_{+0.4}$ | $47.3_{+0.8}$ | $9.3_{+0.1}$ | $29.2_{+1.1}$ | $4.7_{+0.3}$ | $0.5_{+0.0}$ | $13.4_{+0.2}$ | $18.6_{+0.0}$ |
| | RLE (Li et al., 2021) | MobileNetV2 | $27.5_{+1.4}$ | $48.1_{+1.7}$ | $16.7_{+0.8}$ | $43.1_{+2.7}$ | $11.1_{+0.8}$ | $2.0_{+0.7}$ | $17.7_{+0.8}$ | $19.5_{+0.9}$ |
| | | ResNet-50 | $28.2_{+1.7}$ | $47.1_{+3.1}$ | $16.3_{+2.5}$ | $41.2_{+6.9}$ | $11.4_{+1.4}$ | $1.3_{+0.8}$ | $17.4_{+2.8}$ | $20.0_{+1.6}$ |
| | | ResNet-101 | $28.2_{+3.5}$ | $46.5_{+4.3}$ | $16.2_{+2.6}$ | $41.1_{+5.7}$ | $10.8_{+2.4}$ | $2.1_{+0.1}$ | $17.3_{+2.8}$ | $20.1_{+2.1}$ |
| | | ResNet-152 | $30.0_{+1.3}$ | $48.4_{+2.2}$ | $18.1_{+2.8}$ | $43.0_{+7.9}$ | $13.5_{+0.6}$ | $1.4_{+0.3}$ | $19.2_{+3.2}$ | $22.3_{+1.1}$ |
| *Heatmap-based* | CPM (Wei et al., 2016) | CPM | $40.7_{+0.2}$ | $60.4_{+0.0}$ | $21.6_{+0.1}$ | $51.0_{+0.4}$ | $17.1_{+0.1}$ | $9.5_{+0.6}$ | $22.4_{+0.1}$ | $25.4_{+0.1}$ |
| | Hourglass (Newell et al., 2016) | Hourglass-4 | $44.6_{+0.5}$ | $60.8_{+0.1}$ | $20.6_{+0.3}$ | $48.9_{+0.1}$ | $16.0_{+0.4}$ | $4.6_{+0.1}$ | $23.7_{+0.6}$ | $28.2_{+0.2}$ |
| | MobileNetV2 (Sandler et al., 2018) | MobileNetV2 | $39.8_{+0.4}$ | $59.4_{+0.4}$ | $19.4_{+0.1}$ | $48.5_{+0.6}$ | $14.3_{+0.8}$ | $2.3_{+0.1}$ | $20.6_{+0.1}$ | $23.2_{+0.1}$ |
| | SimpleBaseline (Xiao et al., 2018) | ResNet-50 | $43.3_{+0.2}$ | $61.7_{+1.2}$ | $22.1_{+0.2}$ | $51.5_{+0.4}$ | $17.7_{+0.2}$ | $3.7_{+0.4}$ | $23.4_{+0.2}$ | $26.3_{+0.1}$ |
| | | ResNet-101 | $42.8_{+0.3}$ | $60.7_{+0.2}$ | $21.7_{+0.1}$ | $52.5_{+0.4}$ | $16.7_{+0.0}$ | $4.3_{+0.2}$ | $23.0_{+0.1}$ | $26.2_{+0.2}$ |
| | | ResNet-152 | $43.9_{+0.4}$ | $61.6_{+0.1}$ | $22.7_{+0.4}$ | $53.4_{+0.6}$ | $18.3_{+0.4}$ | $5.3_{+0.5}$ | $23.9_{+0.4}$ | $27.1_{+0.1}$ |
| | HRNet (Sun et al., 2019) | HRNet-W32 | $48.6_{+0.9}$ | $65.6_{+0.6}$ | $25.9_{+0.4}$ | $58.2_{+0.8}$ | $22.1_{+0.4}$ | $6.1_{+0.4}$ | $27.0_{+0.6}$ | $30.3_{+0.5}$ |
| | | HRNet-W48 | $47.3_{+0.2}$ | $64.5_{+0.2}$ | $25.1_{+0.1}$ | $57.2_{+0.6}$ | $21.0_{+0.1}$ | $6.9_{+0.9}$ | $26.2_{+0.3}$ | $29.6_{+0.1}$ |
| | DarkPose (Zhang et al., 2020) | ResNet-50 | $43.7_{+0.0}$ | $62.1_{+0.6}$ | $22.8_{+0.1}$ | $53.8_{+0.8}$ | $18.8_{+0.6}$ | $3.4_{+0.2}$ | $24.1_{+0.0}$ | $27.1_{+0.1}$ |
| | | ResNet-101 | $43.1_{+0.9}$ | $61.2_{+1.4}$ | $22.1_{+0.3}$ | $52.6_{+0.6}$ | $17.6_{+0.7}$ | $4.0_{+0.4}$ | $23.4_{+0.3}$ | $26.5_{+0.3}$ |
| | | ResNet-152 | $43.5_{+0.3}$ | $61.2_{+0.2}$ | $22.4_{+0.1}$ | $53.2_{+0.1}$ | $17.4_{+0.3}$ | $4.6_{+0.0}$ | $23.7_{+0.1}$ | $26.7_{+0.1}$ |
| | | HRNet-W32 | $48.7_{+0.5}$ | $65.6_{+0.9}$ | $25.7_{+0.4}$ | $58.4_{+0.8}$ | $21.3_{+0.8}$ | $5.6_{+0.4}$ | $26.9_{+0.2}$ | $30.1_{+0.2}$ |
| | | HRNet-W48 | $47.6_{+0.7}$ | $64.5_{+1.0}$ | $25.8_{+0.4}$ | $58.0_{+1.7}$ | $21.5_{+0.3}$ | $6.6_{+0.5}$ | $27.0_{+0.4}$ | $30.2_{+0.5}$ |
| | HRFormer (Yuan et al., 2021) | HRFormer-S | $45.1_{+0.4}$ | $61.4_{+0.4}$ | $23.0_{+0.0}$ | $53.1_{+0.4}$ | $19.7_{+0.2}$ | $5.5_{+1.6}$ | $24.1_{+0.2}$ | $27.1_{+0.1}$ |
| | | HRFormer-B | $46.4_{+0.3}$ | $63.0_{+0.1}$ | $24.1_{+0.6}$ | $55.3_{+1.0}$ | $20.1_{+0.1}$ | $5.2_{+0.4}$ | $25.4_{+0.5}$ | $28.2_{+0.4}$ |
| | HRNet (Sun et al., 2019) | HRNet-W32 | ⊘ | ⊘ | 74.4 | 90.5 | 81.9 | 70.8 | 81.0 | 79.8 |

The train/test partitioning mirrors that of the first track. We use mAP with various thresholds, adhering to the conventions of human pose estimation (Lin et al., 2014). Additionally, we report the Percentage of Correctly estimated Keypoints (PCK) metric (Andriluka et al., 2014; Ng et al., 2022). PCK@$\alpha$ quantifies the fraction of accurately predicted keypoints within a distance threshold defined as $\alpha \times max(height, width)$, derived from the bounding box of the chimpanzee. This metric is widely recognized for its accuracy in body joint localization in both human and animal pose estimation.

**Results** Tab. 4 consolidates these pose estimators' performances on the `ChimpACT` test set. Notably, the heatmap-based DarkPose (Zhang et al., 2020) with an HRNet (Sun et al., 2019) backbone emerges as the top-performing model. This trend aligns with observations in human pose estimation, where heatmap-centric methods (Wei et al., 2016; Xiao et al., 2018; Newell et al., 2016; Sun et al., 2019) predominantly lead the pack, attributed to their robustness against pose and appearance variations. However, the heatmap representation may be less accurate in scenarios where multiple joints are occluded or closely spaced, and it demands heftier computational and memory resources. Conversely, the newer regression-based methods (Li et al., 2021) are computationally leaner but tend to be more susceptible to overfitting and generally lag in performance.

These results underscore that the task of chimpanzee pose estimation is distinct and nuanced, and cannot be seamlessly addressed by merely repurposing human-centric pose estimation methods. We believe there are two primary reasons for this: (i) chimpanzees exhibit unique joint flexibility and a broader range of motion, and (ii) the visual texture and appearance of chimpanzee fur diverge significantly from human skin. These insights emphasize the need for chimpanzee specific pose estimation strategies.

## 4.3 Spatiotemporal action detection

**Setting** We benchmark four representative human action detection baselines on `ChimpACT` using the MMAction2 (Contributors, 2020c) codebase, including ARCN (Sun et al., 2018), LFB (Wu et al., 2019), and SlowFast with its variant SlowOnly (Feichtenhofer et al., 2019). All models undergo training for 20 epochs with a batch size of 32. Convergence is evident from the training curves

Table 5: **Results of spatiotemporal action detection track on `ChimpACT` test set.** The row highlighted in light blue is the performance reference on the human action dataset AVA (Gu et al., 2018). − denotes not applicable. "*w*. NL/Max/Avg LFB" denotes using non-local, max, or average LFB module. "*w*. Ctx" indicates using both the RoI feature and the global pooled feature for classification. "mAP," "mAP$_L$," "mAP$_O$," "mAP$_S$," and "mAP$_o$" represent the overall mAP and mAP for *L*ocomotion, *O*bject interaction, *S*ocial interaction, and *o*thers.

| Method | Frame sampling | Module | mAP | mAP$_L$ | mAP$_O$ | mAP$_S$ | mAP$_o$ |
|---|---|---|---|---|---|---|---|
| ACRN (Sun et al., 2018) | $8 \times 8 \times 1$ | | $24.4_{+0.5}$ | $58.7_{+0.7}$ | $33.8_{+1.7}$ | $14.7_{+0.4}$ | $0.0_{+0.0}$ |
| | $4 \times 16 \times 1$ | | $23.9_{+1.3}$ | $57.8_{+0.4}$ | $35.0_{+4.0}$ | $13.8_{+1.6}$ | $0.0_{+0.0}$ |
| LFB (Wu et al., 2019) | $4 \times 16 \times 1$ | *w*. NL LFB | $22.0_{+0.9}$ | $50.1_{+0.8}$ | $32.3_{+0.9}$ | $13.5_{+1.6}$ | $0.6_{+0.1}$ |
| | $4 \times 16 \times 1$ | *w*. Max LFB | $23.2_{+0.7}$ | $45.0_{+1.5}$ | $31.2_{+0.8}$ | $17.7_{+1.4}$ | $0.5_{+0.0}$ |
| | $4 \times 16 \times 1$ | *w*. Avg LFB | $21.3_{+1.6}$ | $45.0_{+3.6}$ | $29.8_{+1.1}$ | $14.7_{+2.6}$ | $0.5_{+0.0}$ |
| SlowOnly (Feichtenhofer et al., 2019) | $8 \times 8 \times 1$ | | $20.9_{+1.9}$ | $48.1_{+7.0}$ | $36.2_{+2.8}$ | $11.5_{+1.0}$ | $0.0_{+0.1}$ |
| | $4 \times 16 \times 1$ | | $19.2_{+1.1}$ | $47.0_{+2.5}$ | $28.3_{+2.5}$ | $11.0_{+1.2}$ | $0.0_{+0.1}$ |
| | $8 \times 8 \times 1$ | *w*. Ctx | $22.3_{+1.9}$ | $52.3_{+3.2}$ | $31.2_{+1.3}$ | $13.8_{+2.4}$ | $0.1_{+0.1}$ |
| | $4 \times 16 \times 1$ | *w*. Ctx | $21.4_{+0.9}$ | $47.6_{+2.0}$ | $33.0_{+1.2}$ | $13.2_{+2.2}$ | $0.2_{+0.1}$ |
| SlowFast (Feichtenhofer et al., 2019) | $8 \times 8 \times 1$ | | $21.9_{+1.0}$ | $53.0_{+0.7}$ | $30.6_{+2.2}$ | $12.9_{+1.2}$ | $0.0_{+0.1}$ |
| | $4 \times 16 \times 1$ | | $22.0_{+0.8}$ | $52.9_{+2.3}$ | $33.1_{+2.3}$ | $12.6_{+0.9}$ | $0.0_{+0.0}$ |
| | $8 \times 8 \times 1$ | *w*. Ctx | $24.3_{+0.6}$ | $56.8_{+1.6}$ | $31.5_{+2.0}$ | $15.6_{+0.8}$ | $0.1_{+0.1}$ |
| | $4 \times 16 \times 1$ | *w*. Ctx | $24.1_{+0.9}$ | $56.6_{+2.0}$ | $34.7_{+2.7}$ | $14.6_{+0.4}$ | $0.1_{+0.1}$ |
| SlowFast (Feichtenhofer et al., 2019) | $8 \times 8 \times 1$ | | 25.8 | − | − | − | − |

in Fig. A2c. We maintain consistent optimizers and learning rates as in official implementations. Ground-truth bounding boxes for each chimpanzee are provided during both training and testing, as per Tang et al. (2020). Please refer to Appx. C for further details on ablative modules.

We adopt the same train-test split as previous tracks. Performance is gauged using mAP across 23 action classes, as per standard (Feichtenhofer et al., 2019; Tang et al., 2020). Additionally, we evaluate the mAP within the four behavioral types separately.

**Results**   Tab. 5 summarizes the action detection algorithms' performances on the `ChimpACT` test set. The overall mAP aligns with results on human action datasets, underscoring the feasibility of automated action detection for video coding and further analyses. Locomotion behaviors achieve a notably higher mAP, likely due to their solitary nature and distinct patterns. Conversely, Conversely, the "others" category registers the lowest mAP, attributed to its limited data—comprising just 0.14% of action instances across two fine-grained classes. This imbalance suggests the potential benefit of few-shot learning methods in the future. The results highlight both the promise and areas for improvement in the dataset, positioning it as a valuable platform for advancing spatiotemporal action detection algorithms. We anticipate that `ChimpACT` will further studies into the social dynamics of non-human primates in semi-naturalistic environments.

## 5   Conclusion

In this work, we introduced `ChimpACT`, a novel longitudinal video dataset capturing the intricate behaviors of group-living chimpanzees, focusing on the juvenile chimpanzee, Azibo. Our meticulous annotations and diverse social interactions within the dataset offer a unique view into the world of our closest evolutionary relatives. Through comprehensive experiments, we underscored the challenges and nuances of applying human-centric computer vision algorithms to the distinct behaviors and interactions of chimpanzees. The dataset's depth, combined with its long-tail distribution, not only emphasizes its significance but also paves the way for interdisciplinary research bridging primatology, comparative psychology, computer vision, and machine learning. By making this resource available, our aspiration is to catalyze advancements in video understanding, inspire the research community to craft specialized techniques for non-human primates and deepen our collective insights into their intricate social fabric and dynamics.

**Limitation and future work**   `ChimpACT` is based on captive chimpanzees living in a semi-natural environment, limiting the observable range of behaviors. Natural foraging, responses to predators, and intergroup encounters are absent. Focusing on Azibo overrepresents certain individuals and underrepresents others, limiting the assessment of the full social network. Nevertheless, we plan to contribute more data and labels to create a larger and more comprehensive chimpanzee dataset.

# Acknowledgement

The authors would like to thank the Wolfgang Köhler Primate Research Center, BasicFinder CO., Ltd., and Keyue Zhang for annotations and quality check, Zihao Yin for discussions and preliminary experiments on the chimpanzee detection models, Guangyuan Jiang and Yuyang Li for their technical support on the GPU cluster, and NVIDIA for their generous support of GPUs and hardware. X. Ma, J. Su, W. Zhu, Y. Zhu, and Y. Wang are supported in part by the National Key R&D Program of China (2022ZD0114900), and Y. Zhu is supported in part by the Beijing Nova Program and the National Comprehensive Experimental Base for Governance of Intelligent Society, Wuhan East Lake High-Tech Development Zone.

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

# A    Additional details on `ChimpACT`

## A.1    Ethogram

We detail the ethogram definition in Tab. A1, which systematically describes the daily behaviors of chimpanzees.

Table A1: **The ethogram used for the `ChimpACT` dataset.**

| category | definition | subcategory | subcategory definition |
|---|---|---|---|
| locomotion | patterns of self-initiated movement of an individual | 0. moving | moving horizontally, *e.g.*, walking, running |
| | | 1. climbing | moving vertically, *e.g.*, climbing up or down a structure |
| | | 2. resting | remaining stationary, *e.g.*, standing, sitting, or lying |
| | | 3. sleeping | resting and keeping eyes closed |
| object interaction | direct physical interactions with inanimate stationary or movable objects by hands, feet or mouth | 4. solitary object playing | non-social and non-goal-directed object interaction and exploration |
| | | 5. eating | consuming and processing food |
| | | 6. manipulating object | manipulation of any kind of inanimate object excluding eating |
| social interaction | at least two chimpanzees are interacting in differentiated roles: with one individual initiating the social behavior (initiator) and one individual receiving the social behavior (recipient) | 7. grooming | a chimpanzee, the groomer, is cleaning the fur, head, hand, feet, or genitals of another chimpanzee, usually using their hands and/or mouth |
| | | 8. being groomed | one chimpanzee, the groomee, is getting their skin or fur cleaned by another chimpanzee |
| | | 9. aggressing | a chimpanzee is showing agonistic behavior towards another chimpanzee. This can range from charging and chasing another chimpanzee to direct physical contact such as slapping, hitting, and biting |
| | | 10. embracing | a chimpanzee is embracing another chimpanzee with their arms, not to be confused with carrying |
| | | 11. begging | a chimpanzee is requesting food or another object from another chimpanzee, oftentimes by extending their arm, reaching, or using an open palm begging gesture |
| | | 12. being begged from | a chimpanzee is requested food or another object by another chimpanzee |
| | | 13. taking object | taking an object from the possession of another chimpanzee, the transfer might be resisted or not |
| | | 14. losing object | the possession is taken by another chimpanzee |
| | | 15. carrying | a chimpanzee (usually an adult) carries another chimpanzee (usually an infant or juvenile) on the back, front, side, arm, or leg for more than 2 steps |
| | | 16. being carried | a chimpanzee (usually an infant or juvenile) is carried by another chimpanzee (usually an adult) on the back, front, side, arm, or leg for more than 2 steps. |
| | | 17. nursing | a female chimpanzee is nursing (breastfed, *i.e.*, making physical contact with the nipple) an infant/juvenile |
| | | 18. being nursed | an infant/juvenile is being nursed (breastfed, *i.e.*, making physical contact with the nipple) by a female chimpanzee |
| | | 19. playing | a chimpanzee is physically interacting with another individual in a friendly, teasing, or mock fighting way (*e.g.*, play fighting and other behaviors) |
| | | 20. touching | a chimpanzee makes body contact with another chimpanzee (*e.g.*, holding hands) and it does not fit with any of the other social interaction categories described above |
| others | other behaviors | 21. erection | a male chimpanzee has an erect penis |
| | | 22. displaying | a male chimpanzee, usually with puffed up hair (piloerection) and an erection, performs a dominance display, which includes walking with a swagger, swinging their arms to the sides, and making calls with increasing amplitude, commonly ending by stomping against or slapping objects. Displays can be directed at another chimpanzee or be undirected |

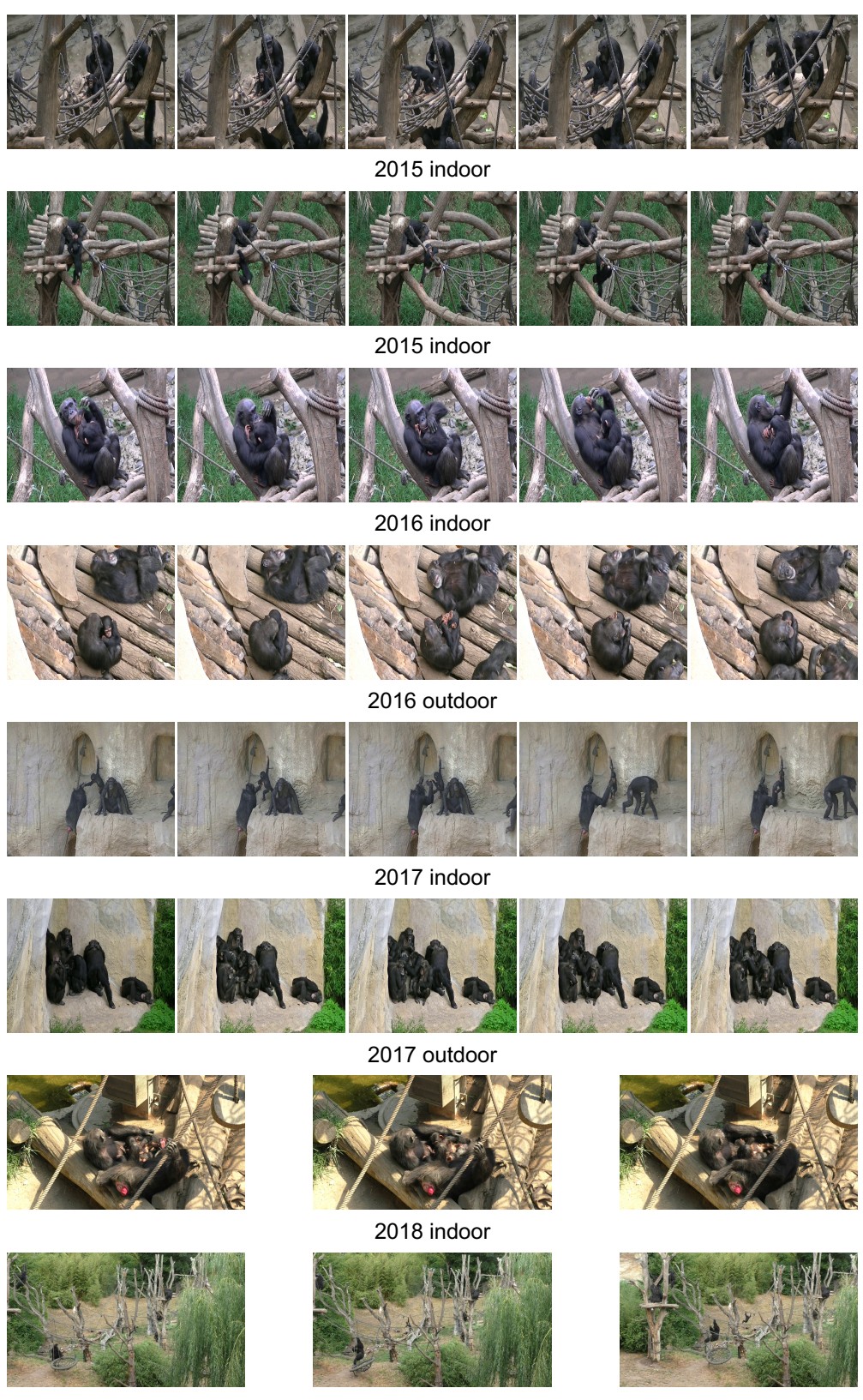

Figure A1: **Example frames from the `ChimpACT` dataset.** `ChimpACT` possesses rich social interactions of the complex everyday life of group-living chimpanzees and contains several environmental enrichment.

## A.2 Dataset details

**Collection and organization**   405 hours of video footage of the Leipzig A-group chimpanzees were collected between 2015 and 2018. To create a representative sample of the footage, 163 video clips were selected, with 15, 35, 86, and 27 clips taken from each year. These video clips cover the four seasons. Each clip is 1000 frames long, with only 3 clips being shorter than 1000 frames. Visual examples from six clips, featuring both indoor and outdoor enclosures, are shown in Fig. A1. The dataset covers a diverse range of physical scenarios, camera views, and social behaviors, as demonstrated in these examples. For instance, in the third row of the figure, an adult chimpanzee is shown grooming an infant chimpanzee in her arms, while later on, the same infant is nursed.

**Annotation process and quality**   The annotation process was conducted using BasicFinder CO., Ltd.'s private labeling platform, which involved a team of 15 annotators and 2 managers. Prior to commencing the annotation work, our team developed comprehensive guidelines that explicitly outlined the requirements for labeling. These guidelines covered several aspects, including:

(i) Assigning a bounding box for each chimpanzee in the image. (ii) Specifying the visibility of the bounding boxes. (iii) Assigning tracking IDs to each bounding box for tracking purposes. (iv) Localizing 2D keypoints within each bounding box. (v) Indicating the visibility of each 2D keypoint. (vi) Assigning behavior labels for each bounding box.

To ensure that the annotators followed these guidelines accurately, the project managers provided training based on the guidelines. Following the training, the annotators performed a trial annotation on a small dataset. We actively sought feedback from the annotators during this phase, which allowed us to address any issues and make necessary improvements. We conducted a thorough review of the trial annotations to verify that the quality met our standards.

During the trial labeling phase, we reached out to three labeling companies and ultimately selected BasicFinder CO., Ltd. based on their exceptional labeling quality. It is worth noting that BasicFinder CO., Ltd. has previously led the annotation efforts for the BDD100K (Yu et al., 2020) dataset, which is a substantial dataset used for autonomous driving purposes. This experience demonstrates their ability to maintain high annotation standards for complex and extensive datasets. Consequently, their involvement improves the reliability of our `ChimpACT` dataset annotations as well.

Once we were confident in the quality of the trial annotations, we proceeded with the large-scale annotation process. To manage the annotations efficiently, each video clip was designated as an annotation task, and our managers assigned these tasks to individual annotators using BasicFinder CO., Ltd.'s platform, ensuring that there was no overlap in assignments. BasicFinder CO., Ltd. has implemented rigorous quality management practices throughout the annotation process. These practices include a customized workflow, complete job traceability, precise performance tracking, multiple levels of auditing, and scientific personnel management. By adhering to these practices, we were able to maintain high standards of quality and accuracy while ensuring efficient processing speed. The annotation process followed a sequential workflow of execution, review, and quality control. Experienced annotators were responsible for executing the annotations, while the manager, as well as our team, conducted thorough reviews and quality control checks. Any annotations that did not meet the required standards were sent back to the annotators for corrections. The quality control phase involved a comprehensive review and verification of all data by both the managers and our own team, ensuring the integrity and accuracy of the annotations. Once all the data had been confirmed to meet our standards of quality, we concluded the annotation process.

More specifically, to label chimpanzee identities, annotators only needed to assign a tracking ID to each chimpanzee, which was then reviewed by the primatologist in our team, who assigned the apes' names based on his knowledge of the observed Leipzig A-group chimpanzees. The process of localizing 2D keypoints within each bounding box and assigning behavior labels for each chimpanzee presented bigger challenges than other tasks. To overcome these challenges, we implemented several measures to ensure accuracy and consistency. For the labeling of 2D keypoints, we provided detailed instructions accompanied by visual illustrations, aiming to provide clear guidelines for annotators to precisely identify and mark the keypoints. For labeling of behaviors, we supplied example videos showcasing different chimpanzee behaviors, created by our team's experienced primatologists. These videos served as valuable references, enabling annotators to accurately assign behavior labels based on observed actions. Throughout the annotation process, the primatologists actively participated, offering their expertise and providing valuable feedback to ensure the annotations aligned with

scientific standards. Finally, the behavioral primatologists in our team manually reviewed all labeled frames to ensure data reliability. These measures and the involvement of the primatologists were instrumental in enhancing the overall quality and reliability of the annotations.

For more information on the dataset, including pre-processing scripts, and visualized annotations, please refer to our project website.

# B   Discussion on `ChimpACT`

**Intended uses**   The `ChimpACT` dataset is a versatile resource that can be used for studying algorithms for chimpanzee detection, tracking, identification, pose estimation, and spatiotemporal action detection. Therefore, the dataset is both relevant for questions in computer vision and primate behavior. In the context of computer vision, it lends itself to other research topics, including but not limited to pose tracking, few-shot learning, weakly-supervised learning, and transfer learning. Considering primate behavior, the dataset shares numerous features with other video data commonly collected with captive and wild chimpanzee populations. This makes it an ideal resource for fine-grained investigations of social (*e.g*., grooming, nursing, aggression) and nonsocial (*e.g*., locomotion, object interactions) chimpanzee behaviors. We strongly encourage researchers to utilize our dataset solely for research purposes that promote animal welfare and conservation. We firmly discourage any use of the dataset for harmful activities such as poaching, hunting or any other exploitation of primates. It is crucial for researchers to approach the data with a focus on positive societal impacts and to refrain from any potential negative consequences.

**Ethics**   The `ChimpACT` dataset raises no ethical concerns regarding the privacy information of human subjects, as it solely focuses on chimpanzees. Studying the social behavior of chimpanzees provides an ethical and efficient means to explore aspects of human sociality due to our phylogenetic proximity. By analyzing their behaviors, we can gain insights into the evolution of human social behavior and potentially contribute to both the scientific and ethical understanding of the human condition. The ethics committee of the Wolfgang Köhler Primate Research Center approved the observational data collection for this project.

**Maintenance, distribution, and license**   The `ChimpACT` dataset will be maintained by the authors and made publicly available with a total of 160,500 frames (around 2 hours) on our project website. The `ChimpACT` dataset will be distributed under the CC BY-NC 4.0 license.

**Wage paid to annotators**   We collaborated with BasicFinder CO., Ltd. for the annotation process. The labeling was carried out by 15 annotators, and they were offered a fair wage as per the prearranged contract. The total expenditure for the labeling process was approximately 70,000 RMB.

# C   Experiments

We trained all the models with officially-used training configurations for each of the three tracks. Please refer to the code implementation on our Github for details. Although we trained the models for different epochs in experiments conducted on different tracks, these choices were made based on conventional practices. Based on the training loss curves provided in Figs. A2a and A2c, it can be observed that all tracking and spatiotemporal action detection methods have reached convergence within the chosen training epochs. To assess the potential overfitting of the pose estimation models, we have included the validation curve on the AP metric in Fig. A2b. The validation curve demonstrates the performance of the pose models on the validation set, which indicates that the pose estimation models are not exhibiting signs of overfitting. Therefore, based on the training loss curves and the validation curve, it can be concluded that the chosen training epochs are appropriate for both tracking and pose estimation methods.

## C.1   Detection, tracking, and ReID

We partitioned the dataset of 163 videos into three sets: 127 videos for training, 17 for validation, and 19 for testing. Of note, all individual chimpanzees are present in both the training and testing sets. In the test set, there are 12 and 7 videos for indoor and outdoor scenes, respectively.

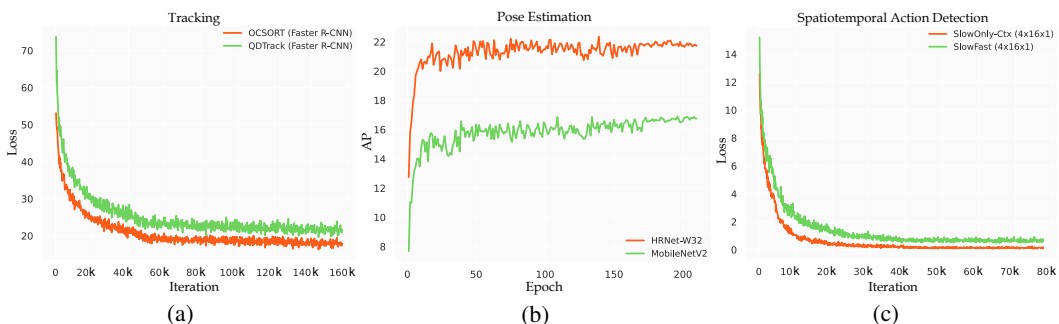

Figure A2: **Training or validation curves on three tracks of example methods.** (a) Training loss curve of example tracking methods. The training iterations correspond to 10 epochs. (b) Validation curve on the AP metric of example pose estimation methods. (c) Training loss curve of example spatiotemporal action detection methods. The training iterations correspond to 20 epochs.

For the evaluation metrics, MOTA (Multiple Object Tracking Accuracy) takes into account FP (False Positives), FN (False Negatives), and IDs (IDentity switches). Usually, FP and FN are larger than IDs; therefore, MOTA mainly assesses the detection performance. IDF1 evaluates the ability to preserve subject identities to assess identification association performance. HOTA (Higher Order Tracking Accuracy) is a recently proposed metric that considers accurate detection, association, and localization equally important, and balances their effects explicitly.

**Results** We additionally evaluated the performance on the **indoor** and **outdoor** test set in Tabs. A2 and A3, respectively. Notably, the results indicate that these approaches achieve consistently better performance on the indoor test set compared to the outdoor test set. This may be attributed to the greater complexity of outdoor scenarios and the presence of varying camera views, which can significantly increase the difficulty of detecting and tracking chimpanzees. Furthermore, the presence of occlusions, similar appearances, and other environmental factors can further exacerbate the challenges of chimpanzee tracking in outdoor settings.

Table A2: **Results of the detection, tracking, and ReID track on `ChimpACT` *indoor* test set.**

| Method | Detector | ReID | HOTA ↑ | MOTA ↑ | MOTP ↑ | IDF1 ↑ | mAP ↑ | FP ↓ | FN ↓ | IDs ↓ |
|---|---|---|---|---|---|---|---|---|---|---|
| **SORT** (Bewley et al., 2016) | Faster R-CNN | ResNet-50 | 49.1 | 52.7 | 21.0 | 49.2 | 76.2 | 8275 | 9396 | 731 |
| | YOLOX | | 41.3 | 46.7 | 18.9 | 38.4 | 77.2 | 6440 | 13163 | 1105 |
| **DeepSORT** (Wojke et al., 2017) | Faster R-CNN | ResNet-50 | 53.2 | 51.6 | 21.0 | 58.3 | 76.2 | 8277 | 9398 | 1144 |
| | YOLOX | | 43.3 | 46.8 | 18.9 | 40.8 | 77.2 | 6440 | 13163 | 1092 |
| Tracktor (Bergmann et al., 2019) | Faster R-CNN | ResNet-50 | 53.6 | 54.5 | 20.6 | 58.3 | 76.2 | 6575 | 10966 | 146 |
| QDTrack (Pang et al., 2021) | Faster R-CNN | – | 53.6 | 53.6 | 20.9 | 58.5 | 76.7 | 8121 | 9591 | 332 |
| ByteTrack (Zhang et al., 2022) | Faster R-CNN | – | 48.8 | 38.9 | 22.1 | 52.3 | 72.7 | 11599 | 11799 | 372 |
| | YOLOX | – | 51.0 | 48.0 | 17.7 | 55.6 | 76.2 | 5080 | 14893 | 245 |
| OC-SORT (Cao et al., 2023) | Faster R-CNN | – | 48.6 | 40.5 | 21.6 | 52.5 | 71.8 | 10022 | 12693 | 431 |
| | YOLOX | – | 49.8 | 47.9 | 19.3 | 53.6 | 75.9 | 7550 | 12292 | 422 |

Table A3: **Results of the detection, tracking, and ReID track on `ChimpACT` *outdoor* test set.**

| Method | Detector | ReID | HOTA ↑ | MOTA ↑ | MOTP ↑ | IDF1 ↑ | mAP ↑ | FP ↓ | FN ↓ | IDs ↓ |
|---|---|---|---|---|---|---|---|---|---|---|
| **SORT** (Bewley et al., 2016) | Faster R-CNN | ResNet-50 | 31.3 | 43.1 | 25.2 | 35.0 | 63.3 | 3288 | 8142 | 422 |
| | YOLOX | | 34.8 | 31.9 | 22.9 | 35.0 | 61.5 | 3649 | 9786 | 751 |
| **DeepSORT** (Wojke et al., 2017) | Faster R-CNN | ResNet-50 | 39.4 | 41.7 | 25.2 | 47.8 | 63.3 | 3280 | 8134 | 726 |
| | YOLOX | | 36.6 | 31.8 | 22.9 | 37.0 | 61.5 | 3649 | 9786 | 788 |
| Tracktor (Bergmann et al., 2019) | Faster R-CNN | ResNet-50 | 38.8 | 42.5 | 24.5 | 45.0 | 63.3 | 2734 | 9146 | 94 |
| QDTrack (Pang et al., 2021) | Faster R-CNN | – | 40.0 | 50.5 | 27.1 | 49.6 | 73.3 | 3705 | 6067 | 534 |
| ByteTrack (Zhang et al., 2022) | Faster R-CNN | – | 32.5 | 32.7 | 30.1 | 42.9 | 62.1 | 5346 | 8375 | 312 |
| | YOLOX | – | 44.2 | 37.5 | 23.1 | 51.5 | 60.4 | 1842 | 11042 | 139 |
| OC-SORT (Cao et al., 2023) | Faster R-CNN | – | 28.3 | 27.4 | 29.8 | 39.6 | 60.5 | 5342 | 9341 | 448 |
| | YOLOX | – | 42.7 | 31.6 | 22.6 | 47.8 | 60.5 | 4298 | 9695 | 252 |

We visualize the tracking results in Figs. A3 and A4, with the ground-truth bounding boxes and chimpanzee identities shown in the last row. We visualized the confidence scores of the estimated bounding boxes and their associated IDs in each frame obtained by the evaluated methods. It is worth noting that we do not require individual identification of each chimpanzee, but rather assign the same ID to the same animal across frames, following the common practice in multi-human tracking (Milan

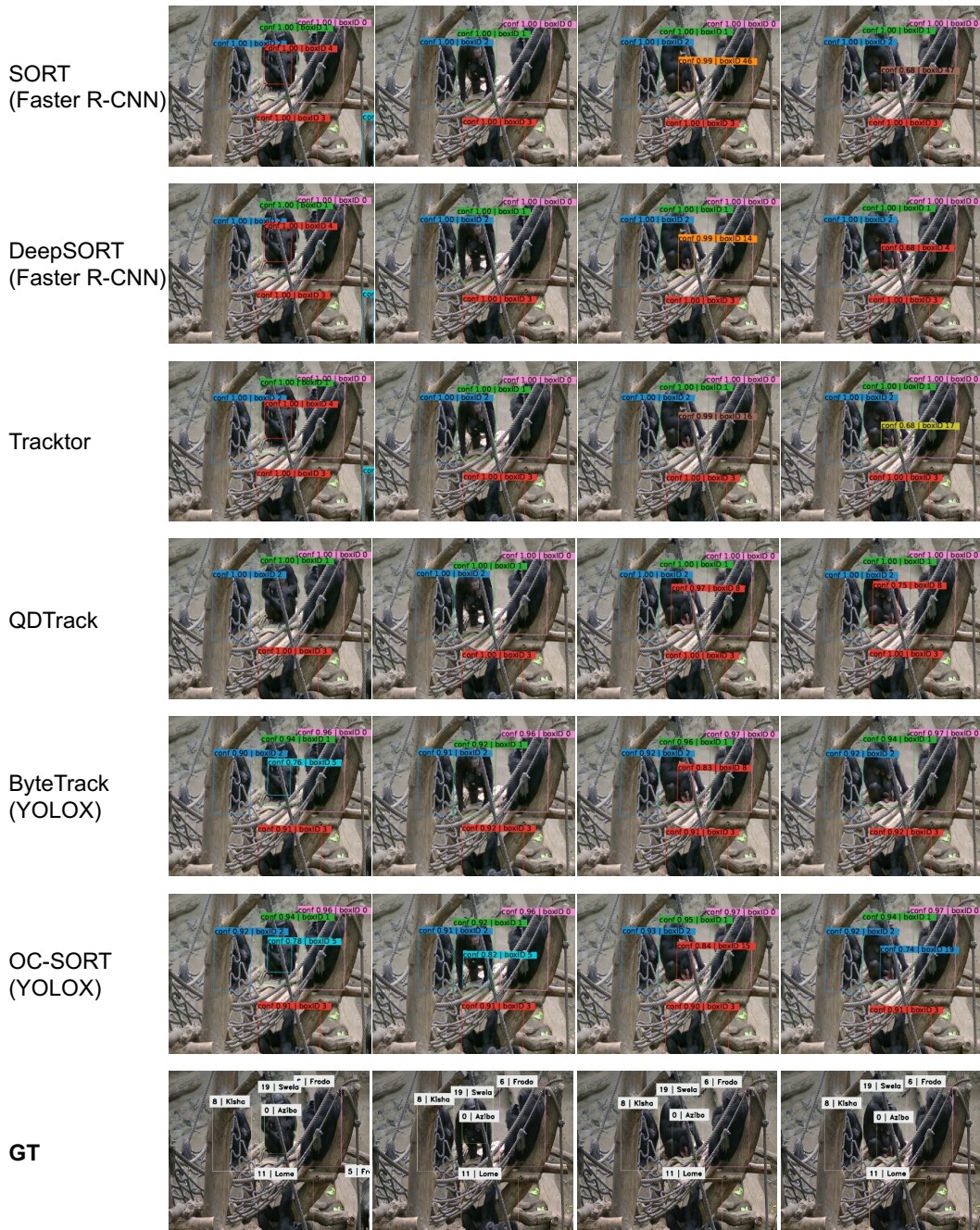

Figure A3: **Qualitative results of representative methods on the `ChimpACT` test set on the tracking task.** For each method, we visualize the estimated confidence score ("conf") and the associated IDs ("boxID") of each bounding box in each frame. The ground-truth bounding boxes and chimpanzee names are shown in the last row, and we add a number left to the name to make it easier to track. Please zoom in for details.

et al., 2016). The estimated box ID is therefore used solely for evaluating the tracking performance. We observed that the evaluated methods performed well in scenarios with minimal occlusion, but struggled to detect and associate the same individual chimpanzee when heavy occlusion occurred. For instance, in Fig. A3, the infant chimpanzee's bounding box is lost in some frames, and its identity is erroneously switched later due to heavy occlusion. This is a challenging task in chimpanzee detection and tracking, as occlusions frequently occur in group-living habitats. Please refer to the supplementary video for more experimental results. In conclusion, the experimental results reveal the limitations of existing methods for chimpanzee detection and tracking, underscoring the need for more robust algorithms to be developed. We believe that our dataset can make a valuable contribution

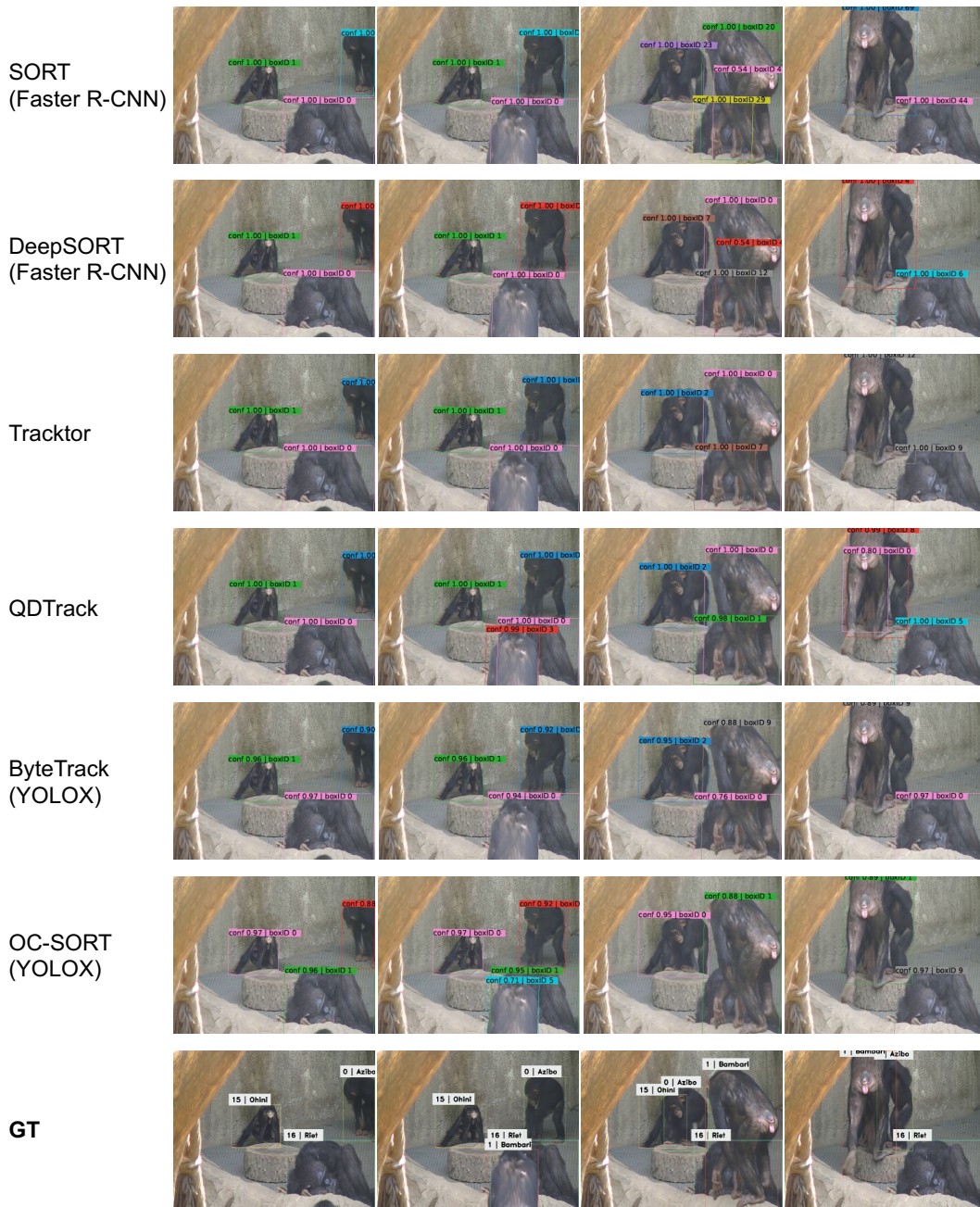

Figure A4: **More qualitative results of representative methods on the `ChimpACT` test set on the tracking task.** For each method, we visualize the estimated confidence score ("conf") and the associated IDs ("boxID") of each bounding box in each frame. The ground-truth bounding boxes and chimpanzee names are shown in the last row, and we add a number left to the name to make it easier to track. Please zoom in for details.

to the advancement of this field, by providing a challenging benchmark for evaluating and comparing different methods.

## C.2 Pose estimation

We followed the partition of the dataset as the first track to train and evaluate the methods.

**Results**    We report the PCK@0.1 for the 16 keypoints in Tab. A4. The results reveal that the keypoints on the face, such as the eyes and lips, exhibited better estimation compared to the arms and legs. This could be attributed to the fact that eyes and lips have more distinctive visual patterns than

Table A4: **Results of the pose estimation track for each keypoint on `ChimpACT` test set.** We report PCK@0.1 metric. We abbreviate the keypoint names. Please refer to Sec. 3.3 for the keypoint definition.

| | Method | Backbone | 0.hip | 1.rknee | 2.rankle | 3.lknee | 4.lankle | 5.neck | 6.ulip | 7.llip | 8.reye | 9.leye | 10.rshoul | 11.relbow | 12.rwrist | 13.lshoul | 14.lelbow | 15.lwrist |
|---|---|---|---|---|---|---|---|---|---|---|---|---|---|---|---|---|---|---|
| Regression | SimpleBaseline (Xiao et al., 2018) | ResNet-50 | 51.1 | 45.8 | 52.3 | 44.7 | 48.8 | 56.4 | 76.2 | 77.9 | 85.7 | 85.2 | 54.7 | 46.1 | 29.2 | 60.5 | 48.5 | 31.5 |
| | | ResNet-101 | 51.3 | 49.0 | 53.3 | 47.3 | 50.2 | 58.2 | 77.1 | 78.7 | 86.4 | 86.4 | 57.9 | 46.8 | 32.5 | 60.2 | 51.8 | 35.4 |
| | | ResNet-152 | 50.6 | 50.5 | 56.8 | 47.4 | 45.3 | 58.3 | 76.4 | 77.4 | 86.8 | 86.0 | 55.8 | 45.1 | 35.2 | 58.2 | 51.3 | 35.8 |
| | RLE (Li et al., 2021) | MobileNetV2 | 53.1 | 46.9 | 53.8 | 49.0 | 48.7 | 61.4 | 77.1 | 78.7 | 86.3 | 85.1 | 59.2 | 41.6 | 33.5 | 59.0 | 48.2 | 31.9 |
| | | ResNet-50 | 47.7 | 42.6 | 46.6 | 42.7 | 46.2 | 57.7 | 75.9 | 77.4 | 81.4 | 79.3 | 59.0 | 44.0 | 30.3 | 58.9 | 48.5 | 30.5 |
| | | ResNet-101 | 51.9 | 49.4 | 52.8 | 55.4 | 49.1 | 61.0 | 79.5 | 80.4 | 87.2 | 86.6 | 60.3 | 46.4 | 40.2 | 62.0 | 53.6 | 39.0 |
| | | ResNet-152 | 54.1 | 50.2 | 52.8 | 53.1 | 49.3 | 60.6 | 79.8 | 80.7 | 88.0 | 85.4 | 63.1 | 50.7 | 42.5 | 61.1 | 53.5 | 38.6 |
| Heatmap-based | CPM (Wei et al., 2016) | CPM | 61.1 | 65.9 | 71.7 | 59.7 | 68.7 | 67.3 | 85.5 | 87.2 | 91.1 | 90.5 | 67.0 | 60.1 | 59.6 | 67.8 | 66.3 | 53.4 |
| | Hourglass (Newell et al., 2016) | Hourglass-4 | 62.4 | 65.3 | 70.8 | 65.2 | 67.8 | 66.4 | 84.0 | 85.9 | 86.9 | 87.3 | 68.5 | 61.7 | 60.4 | 67.7 | 66.0 | 56.0 |
| | MobileNetV2 (Sandler et al., 2018) | MobileNetV2 | 58.8 | 64.8 | 71.2 | 61.3 | 64.8 | 67.0 | 83.8 | 85.4 | 91.0 | 89.1 | 69.3 | 58.6 | 56.9 | 67.4 | 64.8 | 52.1 |
| | SimpleBaseline (Xiao et al., 2018) | ResNet-50 | 63.2 | 67.9 | 70.7 | 64.4 | 67.5 | 66.8 | 85.1 | 86.3 | 92.8 | 90.5 | 70.6 | 59.1 | 57.7 | 67.6 | 65.0 | 54.4 |
| | | ResNet-101 | 62.0 | 64.6 | 69.6 | 61.4 | 68.4 | 67.4 | 85.1 | 87.4 | 91.9 | 89.6 | 70.1 | 61.2 | 56.3 | 66.7 | 63.5 | 54.1 |
| | | ResNet-152 | 64.5 | 64.6 | 69.2 | 62.5 | 69.9 | 67.3 | 86.5 | 88.5 | 91.1 | 89.7 | 72.4 | 62.4 | 58.5 | 69.9 | 66.0 | 55.3 |
| | HRNet (Sun et al., 2019) | HRNet-W32 | 65.8 | 69.5 | 74.5 | 66.1 | 69.2 | 70.5 | 88.2 | 90.4 | 92.6 | 92.1 | 76.1 | 67.7 | 64.4 | 72.4 | 69.5 | 62.8 |
| | | HRNet-W48 | 61.5 | 69.1 | 74.6 | 65.1 | 70.4 | 70.7 | 87.5 | 88.9 | 93.8 | 92.2 | 75.3 | 64.7 | 61.1 | 72.1 | 70.6 | 58.9 |
| | DarkPose (Zhang et al., 2020) | ResNet-50 | 62.6 | 64.4 | 68.6 | 63.2 | 66.6 | 69.9 | 86.3 | 87.7 | 91.7 | 90.5 | 73.8 | 61.5 | 59.1 | 69.6 | 66.9 | 58.0 |
| | | ResNet-101 | 61.7 | 62.9 | 70.5 | 62.6 | 65.7 | 67.0 | 86.3 | 87.7 | 92.0 | 89.7 | 70.1 | 59.7 | 55.4 | 68.6 | 62.8 | 54.4 |
| | | ResNet-152 | 63.3 | 68.6 | 69.1 | 62.5 | 66.0 | 67.7 | 86.5 | 88.0 | 92.6 | 89.5 | 71.8 | 61.9 | 56.1 | 69.5 | 63.3 | 53.4 |
| | | HRNet-W32 | 63.5 | 67.3 | 74.0 | 67.2 | 71.6 | 70.0 | 88.3 | 89.5 | 93.4 | 92.1 | 75.6 | 65.3 | 64.3 | 73.1 | 69.2 | 62.6 |
| | | HRNet-W48 | 65.9 | 69.7 | 73.5 | 67.1 | 72.8 | 72.0 | 89.6 | 91.3 | 94.5 | 91.8 | 73.3 | 62.6 | 61.2 | 71.6 | 70.8 | 62.8 |
| | HRFormer (Yuan et al., 2021) | HRFormer-S | 63.0 | 66.5 | 70.7 | 64.2 | 68.5 | 67.5 | 84.5 | 85.6 | 91.0 | 89.1 | 71.3 | 61.0 | 59.1 | 68.3 | 64.9 | 56.0 |
| | | HRFormer-B | 61.4 | 67.2 | 71.9 | 66.3 | 70.9 | 67.7 | 84.9 | 86.2 | 93.6 | 90.6 | 71.9 | 66.3 | 62.3 | 70.8 | 67.2 | 58.0 |

limbs, which are often surrounded by heavy fur. Tab. A5 further reports the PCK@0.1 for each action category on the test set. We observe that different action types exhibit variations in pose accuracy, for example, with climbing generally achieving slightly higher accuracy compared to resting in most methods. This observation can be attributed to the higher potential for self-occlusion during resting, as chimpanzees tend to exhibit significant self-occlusion due to their flexible joints. This is evident in the visualized examples in Fig. A5, where (a) and (c) depict resting poses with pronounced self-occlusion. In contrast, during climbing, the body is mostly in an extended state, as shown in (b) and (d). Consequently, the PCK tends to be slightly higher for climbing compared to resting as shown in Tab. A6. To validate this assumption, we further evaluate the performance of all the methods for non-occluded poses in Tab. A7. It is interesting to note that all the methods achieve high PCK accuracy when all the keypoints are visible. This demonstrates their effectiveness in accurately estimating poses when occlusions are minimal or absent.

These observations highlight the unique and intricate nature of chimpanzee pose estimation, which is complicated by their flexible joint articulations and extended range of motion, as well as the dissimilar physical appearances of their fur in comparison to that of humans. Consequently, developing accurate pose estimation algorithms for chimpanzees requires careful consideration and specialized techniques that account for their unique characteristics.

Figs. A6 and A7 present the qualitative results of several models on the `ChimpACT` test split, with the ground-truth poses displayed in the last row. It is promising to observe that directly transferring human pose estimation algorithms to chimpanzees yielded decent performance. However, due to self-occlusions and different physical appearance and joint articulations, these models are susceptible to errors in estimating the positions of limbs, as seen in the misaligned right arm and leg of the young chimpanzee in the first column of Fig. A6 and the third column of Fig. A7.

Table A6: **Results of the pose estimation by HRNet-W32 model.** We report PCK@0.05 and PCK@0.1 metrics.

| No. | Action | PCK@0.05 | PCK@0.1 |
|---|---|---|---|
| (a) | resting | 43.8 | 62.5 |
| (b) | climbing | 81.2 | 93.8 |
| (c) | resting | 68.8 | 93.8 |
| (d) | climbing | 75.0 | 100.0 |

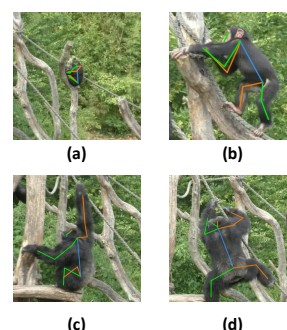

Figure A5: **Visualization of predicted pose by HRNet-W32** (Sun et al., 2019).

A8

Table A5: **Results of the pose estimation track for each action category on `ChimpACT` test set.** We report PCK@0.1 metric. The action category number is consistent with Tab. A1.

| | Method | Backbone | 0 | 1 | 2 | 3 | 4 | 5 | 6 | 7 | 8 | 9 | 10 | 11 | 12 | 15 | 16 | 17 | 18 | 19 | 20 | 21 | 22 |
|---|---|---|---|---|---|---|---|---|---|---|---|---|---|---|---|---|---|---|---|---|---|---|---|
| *Regression* | SimpleBaseline (Xiao et al., 2018) | ResNet-50 | 39.8 | 47.7 | 48.0 | 56.1 | 44.4 | 64.2 | 56.7 | 35.7 | 26.1 | 81.3 | 67.0 | 51.3 | 45.0 | 26.5 | 34.5 | 3.7 | 5.5 | 29.9 | 26.9 | 87.5 | 56.6 |
| | | ResNet-101 | 39.3 | 49.0 | 48.1 | 59.5 | 46.3 | 60.9 | 57.5 | 38.9 | 20.7 | 75.0 | 69.5 | 57.5 | 45.0 | 32.1 | 35.5 | 2.9 | 6.6 | 28.8 | 26.6 | 62.5 | 52.5 |
| | | ResNet-152 | 40.8 | 45.6 | 49.9 | 56.2 | 46.5 | 63.9 | 54.8 | 37.7 | 23.6 | 68.8 | 67.4 | 62.5 | 51.3 | 30.6 | 34.3 | 6.6 | 5.3 | 29.5 | 26.1 | 75.0 | 56.6 |
| | RLE (Li et al., 2021) | MobileNetV2 | 40.8 | 48.0 | 50.0 | 52.9 | 47.1 | 63.5 | 57.7 | 38.4 | 18.1 | 62.5 | 67.6 | 53.8 | 48.8 | 28.8 | 36.5 | 5.7 | 8.5 | 29.4 | 29.8 | 62.5 | 62.5 |
| | | ResNet-50 | 42.0 | 52.1 | 51.5 | 57.6 | 50.0 | 65.2 | 57.8 | 41.0 | 20.2 | 75.0 | 69.0 | 50.0 | 55.0 | 31.9 | 35.4 | 4.6 | 3.2 | 29.0 | 31.5 | 81.3 | 61.3 |
| | | ResNet-101 | 43.3 | 46.4 | 51.8 | 58.3 | 46.6 | 68.0 | 55.8 | 34.1 | 18.7 | 75.0 | 68.5 | 48.8 | 43.8 | 31.9 | 35.2 | 6.0 | 5.6 | 29.7 | 31.6 | 75.0 | 62.6 |
| | | ResNet-152 | 41.4 | 52.9 | 50.7 | 57.2 | 49.3 | 64.2 | 56.3 | 34.1 | 17.7 | 75.0 | 72.5 | 48.8 | 46.3 | 35.6 | 31.8 | 5.4 | 6.1 | 30.2 | 30.1 | 75.0 | 59.8 |
| *Heatmap-based* | CPM (Wei et al., 2016) | CPM | 49.4 | 59.4 | 59.0 | 60.9 | 53.9 | 73.1 | 65.2 | 46.6 | 28.4 | 81.3 | 66.6 | 52.5 | 60.0 | 41.6 | 34.8 | 10.6 | 3.8 | 36.1 | 41.8 | 68.8 | 66.2 |
| | Hourglass (Newell et al., 2016) | Hourglass-4 | 48.1 | 66.5 | 55.3 | 63.2 | 58.5 | 71.9 | 67.4 | 50.3 | 27.8 | 81.3 | 72.8 | 47.5 | 65.0 | 44.0 | 38.2 | 14.1 | 1.8 | 40.6 | 35.3 | 81.3 | 63.6 |
| | MobileNetV2 (Sandler et al., 2018) | MobileNetV2 | 49.8 | 58.4 | 56.1 | 59.3 | 54.8 | 71.2 | 65.1 | 52.8 | 25.8 | 75.0 | 72.8 | 60.0 | 48.8 | 39.8 | 35.8 | 11.7 | 1.5 | 35.0 | 36.2 | 81.3 | 61.4 |
| | SimpleBaseline (Xiao et al., 2018) | ResNet-50 | 52.3 | 60.0 | 57.2 | 60.9 | 56.3 | 73.9 | 66.0 | 53.3 | 25.2 | 81.3 | 72.0 | 62.5 | 65.0 | 45.0 | 35.4 | 8.4 | 2.4 | 39.1 | 37.6 | 75.0 | 65.7 |
| | | ResNet-101 | 52.0 | 60.9 | 57.5 | 60.8 | 56.6 | 71.9 | 66.4 | 52.6 | 28.2 | 93.8 | 72.2 | 71.3 | 61.3 | 42.0 | 34.2 | 6.4 | 1.9 | 39.6 | 36.9 | 68.8 | 67.0 |
| | | ResNet-152 | 51.4 | 60.0 | 57.8 | 60.0 | 57.4 | 71.6 | 66.3 | 55.4 | 27.8 | 81.3 | 79.1 | 58.8 | 52.5 | 45.1 | 32.3 | 5.3 | 0.5 | 38.1 | 38.0 | 87.5 | 67.6 |
| | HRNet (Sun et al., 2019) | HRNet-W32 | 56.7 | 66.1 | 60.8 | 60.9 | 60.2 | 76.3 | 69.3 | 54.8 | 27.2 | 87.5 | 74.8 | 61.3 | 63.8 | 50.6 | 38.7 | 9.1 | 2.4 | 40.2 | 41.4 | 81.3 | 65.6 |
| | | HRNet-W48 | 56.9 | 65.9 | 59.3 | 60.9 | 60.3 | 75.7 | 70.3 | 53.2 | 30.2 | 87.5 | 74.2 | 66.3 | 67.5 | 52.9 | 37.6 | 13.9 | 2.9 | 41.3 | 39.7 | 87.5 | 65.9 |
| | DarkPose (Zhang et al., 2020) | ResNet-50 | 52.1 | 60.9 | 57.5 | 62.1 | 57.4 | 72.6 | 66.1 | 56.0 | 26.8 | 75.0 | 72.9 | 56.3 | 61.3 | 42.1 | 31.7 | 9.6 | 0.4 | 35.3 | 39.0 | 75.0 | 66.4 |
| | | ResNet-101 | 52.6 | 62.6 | 57.6 | 61.4 | 56.1 | 71.8 | 67.7 | 51.7 | 26.3 | 81.3 | 73.4 | 65.0 | 62.5 | 44.0 | 38.2 | 6.3 | 2.6 | 36.7 | 35.7 | 87.5 | 61.1 |
| | | ResNet-152 | 52.6 | 63.3 | 57.8 | 59.4 | 57.9 | 73.2 | 67.9 | 53.0 | 25.7 | 81.3 | 76.3 | 57.5 | 65.0 | 45.0 | 35.0 | 8.7 | 1.7 | 35.9 | 37.1 | 87.5 | 68.3 |
| | | HRNet-W32 | 56.9 | 68.9 | 62.6 | 62.5 | 61.5 | 74.0 | 69.7 | 56.5 | 26.0 | 81.3 | 81.2 | 58.8 | 72.5 | 52.2 | 42.3 | 9.8 | 2.1 | 41.6 | 44.5 | 81.3 | 67.3 |
| | | HRNet-W48 | 57.6 | 67.7 | 60.3 | 59.2 | 60.3 | 73.7 | 69.6 | 56.3 | 28.5 | 93.8 | 77.2 | 53.8 | 67.5 | 52.8 | 36.7 | 4.2 | 1.4 | 39.7 | 40.4 | 75.0 | 63.9 |
| | HRFormer (Yuan et al., 2021) | HRFormer-S | 52.9 | 62.3 | 55.7 | 59.6 | 56.8 | 72.3 | 68.2 | 54.2 | 23.7 | 93.8 | 75.1 | 68.8 | 52.5 | 45.1 | 33.5 | 2.5 | 1.1 | 40.6 | 34.5 | 62.5 | 66.9 |
| | | HRFormer-B | 54.2 | 63.4 | 58.0 | 61.3 | 58.8 | 72.4 | 68.2 | 52.4 | 25.4 | 81.3 | 77.5 | 55.0 | 67.5 | 46.8 | 37.5 | 12.6 | 0.7 | 40.8 | 37.7 | 75.0 | 63.7 |

Table A7: **Results of the pose estimation track for non-occluded poses on `ChimpACT` test set.** We report the PCK metrics. The non-occluded poses denote those with all keypoints visible.

| | Method | Backbone | PCK@0.05 | PCK@0.1 |
|---|---|---|---|---|
| *Regression* | SimpleBaseline (Xiao et al., 2018) | ResNet-50 | 47.6 | 80.6 |
| | | ResNet-101 | 47.2 | 77.9 |
| | | ResNet-152 | 54.5 | 83.0 |
| | RLE (Li et al., 2021) | MobileNetV2 | 47.7 | 82.4 |
| | | ResNet-50 | 52.9 | 82.4 |
| | | ResNet-101 | 28.4 | 55.1 |
| | | ResNet-152 | 60.0 | 85.5 |
| *Heatmap-based* | CPM (Wei et al., 2016) | CPM | 74.0 | 89.4 |
| | Hourglass (Newell et al., 2016) | Hourglass-4 | 77.6 | 88.5 |
| | MobileNetV2 (Sandler et al., 2018) | MobileNetV2 | 67.4 | 89.0 |
| | SimpleBaseline (Xiao et al., 2018) | ResNet-50 | 75.2 | 89.5 |
| | | ResNet-101 | 68.7 | 84.0 |
| | | ResNet-152 | 71.4 | 87.1 |
| | HRNet (Sun et al., 2019) | HRNet-W32 | 77.6 | 92.1 |
| | | HRNet-W48 | 79.4 | 90.2 |
| | DarkPose (Zhang et al., 2020) | ResNet-50 | 74.6 | 87.1 |
| | | ResNet-101 | 74.6 | 88.7 |
| | | ResNet-152 | 73.0 | 89.1 |
| | | HRNet-W32 | 80.7 | 93.0 |
| | | HRNet-W48 | 78.4 | 87.1 |
| | HRFormer (Yuan et al., 2021) | HRFormer-S | 70.9 | 88.5 |
| | | HRFormer-B | 75.6 | 88.4 |

## C.3 Spatiotemporal action detection

We adopted the same dataset partition as the first track. The frame sampling strategy was defined as $T \times I \times N$. We ablated two strategies that continuously sample one frame every $I$ frames and finally get an input clip with $T$ frames by setting $T \neq 1$. $N$ denotes the number of clips which is used only when $T = 1$. For the four representative methods, we ablated different modules. For LFB (Wu et al., 2019), we ablated different ways of the feature bank operator instantiations, by using non-local (NL) blocks (Wang et al., 2018) or average (Avg) or max (Max) pooling. For SlowFast (Feichtenhofer et al., 2019) and the variant SlowOnly, we ablated the context module (Ctx), which indicates that using both the RoI feature and the global pooled feature for the action classification.

**Results** We report the mAP for each model's best configuration on several subcategory behaviors in Tab. A8. The models exhibit better performance in detecting locomotion and solitary object interactions, possibly because these actions are relatively simple and involve less interaction between individuals, making it easier for the model to distinguish between action patterns. However, there is still considerable room for improvement in existing models for action categories with higher levels of interaction, such as social interactions.

We provide qualitative results in Figs. A8 and A9. All methods recognized the playing action of the two chimpanzees in Fig. A8, but incorrectly classified the touching actions as grooming in Fig. A9. These two action patterns exhibit subtle differences that significantly challenge the models

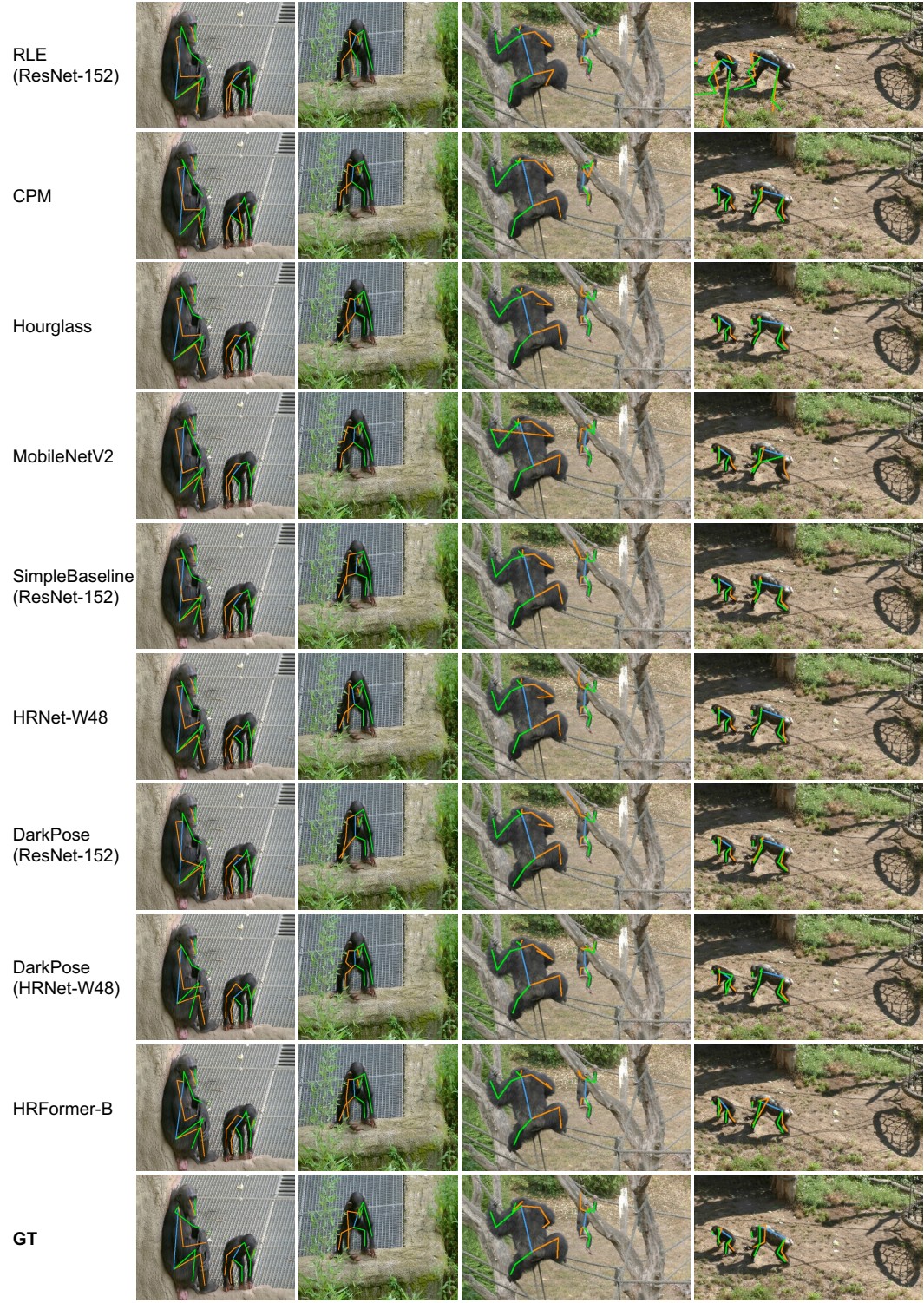

Figure A6: **Qualitative results of representative methods on the `ChimpACT` test set on the pose estimation task.** The ground-truth poses are shown in the last row.

to distinguish them accurately. We recommend referring to the supplementary video for the video results to observe the difference. The challenges of such distinctions highlight the need for stronger algorithms to address these issues effectively.

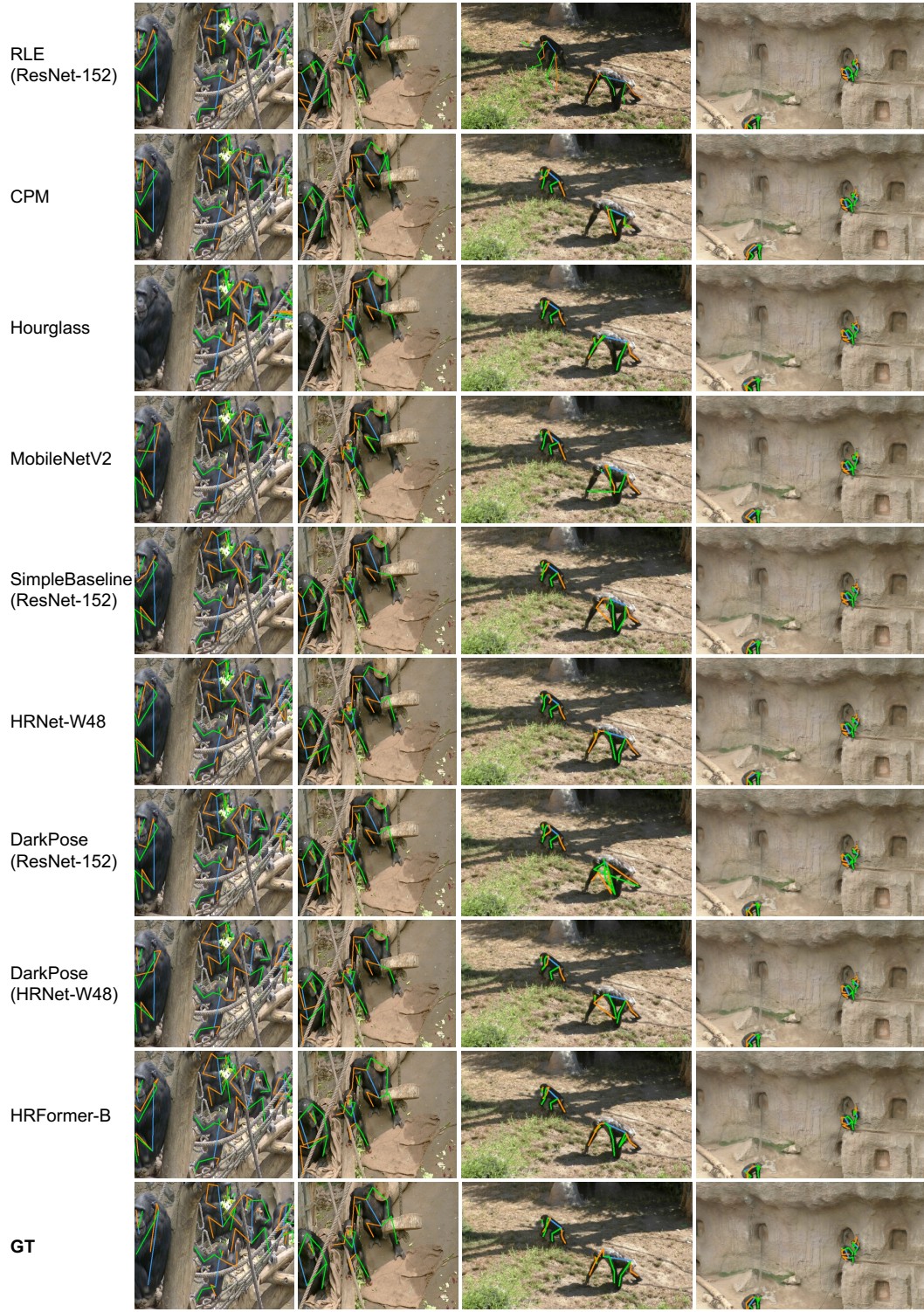

Figure A7: **More qualitative results of representative methods on the `ChimpACT` test set on the pose estimation task.** The ground-truth poses are shown in the last row.

Overall, we hope that our work will inspire further research and development in the area of chimpanzee behavior recognition, with the ultimate goal of improving our understanding of chimpanzee and primate behaviors and ecology.

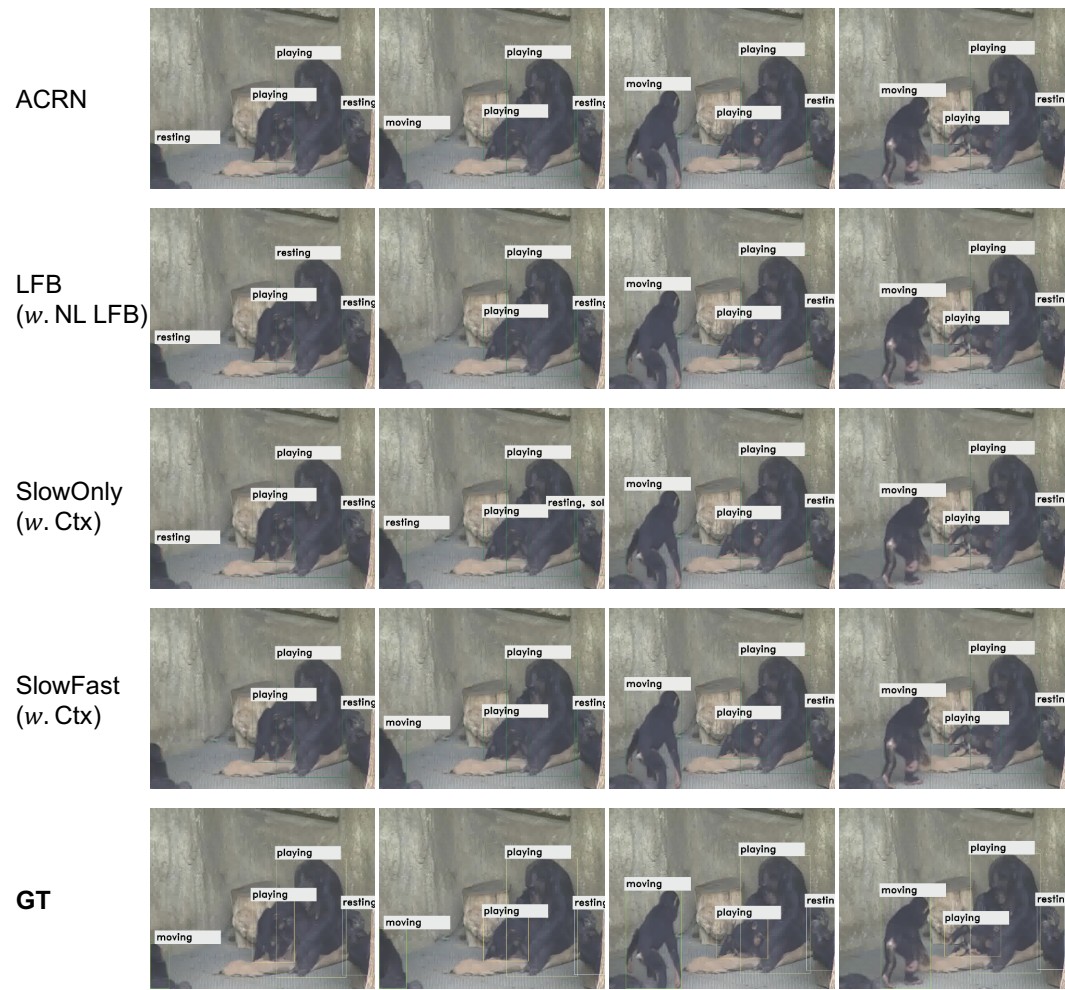

Figure A8: **Qualitative results of representative methods on the `ChimpACT` test set on the spatiotemporal action detection task.** The ground-truth actions are shown in the last row.

Table A8: **Results of spatiotemporal action detection track on `ChimpACT` test set.**

| Method | mAP | moving | climbing | sol. obj. playing | eating | grooming | playing | being begged from | aggressing | being nursed |
|---|---|---|---|---|---|---|---|---|---|---|
| ACRN (Sun et al., 2018) | 24.4 | 60.2 | 23.2 | 38.2 | 54.3 | 7.7 | 42.9 | 0.0 | 0.0 | 4.4 |
| LFB (Wu et al., 2019) | 22.4 | 45.3 | 10.0 | 34.4 | 56.3 | 8.7 | 51.0 | 0.4 | 0.0 | 32.1 |
| SlowOnly (Feichtenhofer et al., 2019) | 24.5 | 56.1 | 31.6 | 41.0 | 45.4 | 10.4 | 43.0 | 0.0 | 0.0 | 7.5 |
| SlowFast (Feichtenhofer et al., 2019) | 24.5 | 60.9 | 37.2 | 47.3 | 35.3 | 10.4 | 49.2 | 0.0 | 0.0 | 7.5 |

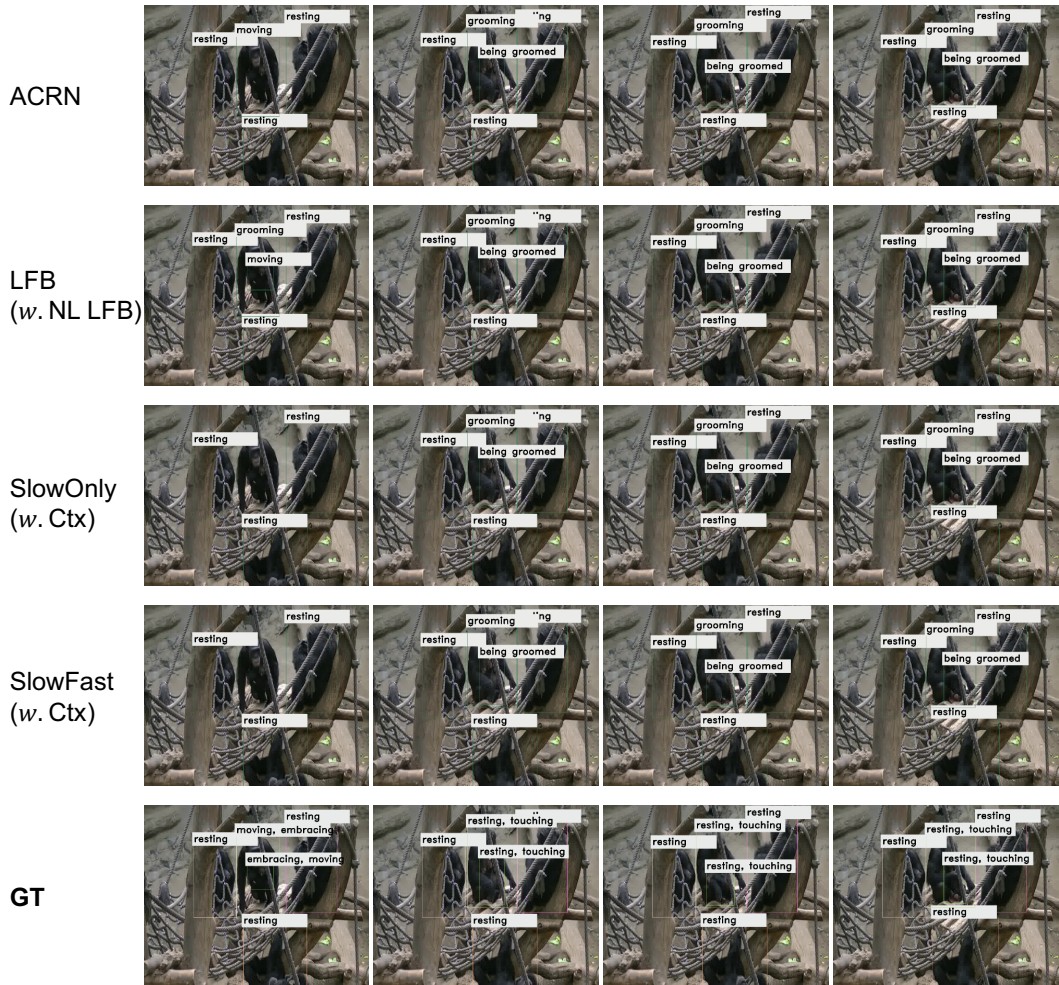

Figure A9: **More qualitative results of representative methods on the `ChimpACT` test set on the spatiotemporal action detection task.** The ground-truth actions are shown in the last row.

# D  Data documentation

We follow the datasheet proposed in Gebru et al. (2021) for documenting our `ChimpACT` and associated benchmarks:

1. **Motivation**

   (a) **For what purpose was the dataset created?**
   This dataset was created to facilitate the study of chimpanzee behaviors, and ultimately advance our understanding of communication and sociality in non-human primates.

   (b) **Who created the dataset and on behalf of which entity?**
   This dataset was created by Xiaoxuan Ma, Stephan P. Kaufhold, Jiajun Su, Wentao Zhu, Jack Terwilliger, Andres Meza, Yixin Zhu, Federico Rossano, and Yizhou Wang. Xiaoxuan Ma, Jiajun Su, Wentao Zhu, Yixin Zhu, and Yizhou Wang are with Peking University. Stephan P. Kaufhold, Jack Terwilliger, Andres Meza, and Federico Rossano are with the University of California, San Diego.

   (c) **Who funded the creation of the dataset?**
   The creation of this dataset was funded by Peking University and the University of California, San Diego.

   (d) **Any other Comments?**
   None.

2. **Composition**

   (a) **What do the instances that comprise the dataset represent?**
   For video data, each instance is a video clip regularized from the raw video. Each instance contains video footage focusing on a group of chimpanzees collected in Leipzig Zoo, Germany. For benchmarking, each instance has rich annotations of chimpanzee identities, poses, and actions. See Sec. 3 and Appx. A.

   (b) **How many instances are there in total?**
   We have 163 video instances in total.

   (c) **Does the dataset contain all possible instances or is it a sample (not necessarily random) of instances from a larger set?**
   No, this is a brand-new dataset.

   (d) **What data does each instance consist of?**
   See Sec. 3 and Appx. A.

   (e) **Is there a label or target associated with each instance?**
   Yes. See Sec. 3 and Appx. A.

   (f) **Is any information missing from individual instances?**
   No.

   (g) **Are relationships between individual instances made explicit?**
   Yes.

   (h) **Are there recommended data splits?**
   Yes, we have separated the whole dataset into training, validation, and test set. See Sec. 4.1, Appx. C and the project website for details.

   (i) **Are there any errors, sources of noise, or redundancies in the dataset?**
   There are almost certainly some errors in video annotations. We did our best to minimize these, but some certainly remain.

   (j) **Is the dataset self-contained, or does it link to or otherwise rely on external resources (*e.g.*, websites, tweets, other datasets)?**
   The dataset is self-contained.

   (k) **Does the dataset contain data that might be considered confidential (*e.g.*, data that is protected by legal privilege or by doctor-patient confidentiality, data that includes the content of individuals' non-public communications)?**
   No.

   (l) **Does the dataset contain data that, if viewed directly, might be offensive, insulting, threatening, or might otherwise cause anxiety?**
   No.

(m) **Does the dataset relate to people?**
No.

(n) **Does the dataset identify any subpopulations (*e.g.*, by age, gender)?**
No.

(o) **Is it possible to identify individuals (*i.e.*, one or more natural persons), either directly or indirectly (*i.e.*, in combination with other data) from the dataset?**
Not applicable. Our dataset only contains chimpanzees.

(p) **Does the dataset contain data that might be considered sensitive in any way (*e.g.*, data that reveals racial or ethnic origins, sexual orientations, religious beliefs, political opinions or union memberships, or locations; financial or health data; biometric or genetic data; forms of government identification, such as social security numbers; criminal history)?**
No.

(q) **Any other comments?**
None.

3. **Collection Process**

(a) **How was the data associated with each instance acquired?**
See Sec. 3.2 and Appx. A for details.

(b) **What mechanisms or procedures were used to collect the data (*e.g.*, hardware apparatus or sensor, manual human curation, software program, software API)?**
We used JVC Everio cameras to collect video footage (Codec H.264). See Sec. 3.2 for details.

(c) **If the dataset is a sample from a larger set, what was the sampling strategy (*e.g.*, deterministic, probabilistic with specific sampling probabilities)?**
See Sec. 3.3 and Appx. A for details.

(d) **Who was involved in the data collection process (*e.g.*, students, crowdworkers, contractors) and how were they compensated (*e.g.*, how much were crowdworkers paid)?**
The video data was collected by the authors. The annotations were performed by the workers in BasicFinder CO., Ltd., and the workers were offered a fair wage as per the prearranged contract. See Sec. 3 and Appx. B for details.

(e) **Over what timeframe was the data collected?**
The data were collected from 2015 to 2018, and labeled in 2022.

(f) **Were any ethical review processes conducted (*e.g.*, by an institutional review board)?**
Not applicable. The `ChimpACT` dataset raises no ethical concerns regarding the privacy information of human subjects, as it solely focuses on chimpanzees.

(g) **Does the dataset relate to people?**
No.

(h) **Did you collect the data from the individuals in question directly, or obtain it via third parties or other sources (*e.g.*, websites)?**
Not applicable.

(i) **Were the individuals in question notified about the data collection?**
Not applicable.

(j) **Did the individuals in question consent to the collection and use of their data?**
Not applicable.

(k) **If consent was obtained, were the consenting individuals provided with a mechanism to revoke their consent in the future or for certain uses?**
Not applicable.

(l) **Has an analysis of the potential impact of the dataset and its use on data subjects (*e.g.*, a data protection impact analysis) been conducted?**
Yes, see Appx. B.

(m) **Any other comments?**
None.

4. **Preprocessing, Cleaning and Labeling**

(a) **Was any preprocessing/cleaning/labeling of the data done (*e.g.*, discretization or bucketing, tokenization, part-of-speech tagging, SIFT feature extraction, removal of instances, processing of missing values)?**
Yes, see Sec. 3.

(b) **Was the "raw" data saved in addition to the preprocessed/cleaned/labeled data (*e.g.*, to support unanticipated future uses)?**
Yes, we provide the raw data on our project website.

(c) **Is the software used to preprocess/clean/label the instances available?**
No. The annotation software is the private labeling platform provided by BasicFinder CO., Ltd. However, existing open-source annotation software such as DeepLabCut (Mathis et al., 2018) could also be used to preprocess/clean/label the instances.

(d) **Any other comments?**
None.

5. **Uses**

(a) **Has the dataset been used for any tasks already?**
No, the dataset is newly proposed by us.

(b) **Is there a repository that links to any or all papers or systems that use the dataset?**
Yes, we provide the link to all related information on our project website.

(c) **What (other) tasks could the dataset be used for?**
This dataset could be used for other research topics, including but not limited to pose tracking, few-shot learning, and transfer learning.

(d) **Is there anything about the composition of the dataset or the way it was collected and preprocessed/cleaned/labeled that might impact future uses?**
We propose to annotate the keyframe every 10 frames for the pose track and action detection track. For tracking track, we label all the frames.

(e) **Are there tasks for which the dataset should not be used?**
The usage of this dataset should be limited to the scope of understanding chimpanzee/non-human primate behaviors.

(f) **Any other comments?**
None.

6. **Distribution**

(a) **Will the dataset be distributed to third parties outside of the entity (*e.g.*, company, institution, organization) on behalf of which the dataset was created?**
Yes, the dataset will be made publicly available.

(b) **How will the dataset be distributed (*e.g.*, tarball on website, API, GitHub)?**
The dataset could be accessed on our project website.

(c) **When will the dataset be distributed?**
The dataset will be released by the end of 2023 on our project website.

(d) **Will the dataset be distributed under a copyright or other intellectual property (IP) license, and/or under applicable terms of use (ToU)?**
We release our benchmark under CC BY-NC 4.0 [1] license.

(e) **Have any third parties imposed IP-based or other restrictions on the data associated with the instances?**
No.

(f) **Do any export controls or other regulatory restrictions apply to the dataset or to individual instances?**
No.

(g) **Any other comments?**
None.

7. **Maintenance**

(a) **Who is supporting/hosting/maintaining the dataset?**
Xiaoxuan Ma is maintaining.

---

[1] https://paperswithcode.com/datasets/license

(b) **How can the owner/curator/manager of the dataset be contacted (*e.g.*, email address)?**
maxiaoxuan@pku.edu.cn

(c) **Is there an erratum?**
Currently, no. As errors are encountered, future versions of the dataset may be released and updated on our website.

(d) **Will the dataset be updated (*e.g.*, to correct labeling errors, add new instances, delete instances')?**
Yes, if applicable.

(e) **If the dataset relates to people, are there applicable limits on the retention of the data associated with the instances (*e.g.*, were individuals in question told that their data would be retained for a fixed period of time and then deleted)?**
Not applicable. The dataset does not relate to people.

(f) **Will older versions of the dataset continue to be supported/hosted/maintained?**
Yes, older versions of the benchmark will be maintained on our website.

(g) **If others want to extend/augment/build on/contribute to the dataset, is there a mechanism for them to do so?**
Yes, please get in touch with us by email.

(h) **Any other comments?**
None.

