# OpenReview forum: "ChimpACT: A Longitudinal Dataset for Understanding Chimpanzee Behaviors"
_NeurIPS.cc/2023/Track/Datasets_and_Benchmarks — NeurIPS 2023 Datasets and Benchmarks Poster_

### Official Review · Reviewer_TrcR · 2023-07-14
**Good contribution but some questions should be addressed.**

**Rating:** 7
**Confidence:** 4
**Correctness:** There are no concern regarding the co…
**Clarity:** The paper is well written and easy to…

**Strengths:**

- The contribution fills a gap as datasets providing behavioral annotations for animals are scarce.
- A large number of baselines are provided for three tasks.
- Good explanation of the dataset statistics.

**Additional Feedback:**

The paper makes a good contribution which will be useful to researchers interested in automatic behavior analysis. However, there remain some concerns regarding baselines and data, which should be addressed.

**Documentation:**

- In the provided material only a limited selection of the data is shown. It would be great to see the variety of backgrounds in the videos, and potentially assess the risk of overfitting to a particular background.
- The authors collected 405 hours of video material over the course of 4 years and selected samples of around 1000 frames length (~40s) resulting in two hours of video material. It would be helpful to mention the criteria used to sample from the complete dataset, in order to assess potential biases.
- Little description of annotation criteria. Are the bounding boxes modal or amodal? How is occlusion handled (e.g if one monkey occludes another, are both bounding boxes retained?)? Since there were multiple annotators: Was inter-annotator reliability evaluated?
- Where do the videos come from? Were they gathered by the authors or provided by Leipzig Zoo or MPI? This is important to judge the contribution of the dataset.

**Ethics:**

No ethical concerns.

**Limitations:**

Limitations are discussed (captivity, focus on one individual).

**Opportunities For Improvement:**

- All recordings are from a single zoo and a single group of chimpanzees. This likely hinders generalization into the wild.
- Line 103: the authors claim to detect and identify at the same time. It is unclear how the baseline multi-object tracking methods are combined with individual identification as most do not support this by default.
- The execution of the baselines could be improved:
    - Bad performance for pose estimation baselines compared to human pose. What is the reason for this?
    - For tracking, all methods are trained for 10 epochs, for pose estimation all models are trained for 210 epoch. There is no explanation for this choice, nor is it clear whether some models are trained well enough (in the case of tracking) or if they are already overfitting (in the case of pose estimation).
    - For spatiotemporal action detection, all methods are from 2019 or older. [1] could be a candidate. I don’t think this is a big issue, but it would be interesting to know why no methods from the more recent years are compared.

Minor:
- Following the development of multi-object tracking field, providing HOTA as the primary metric would make sense.


[1] Arnab et al.: Unified Graph Structured Models for Video Understanding

**Relation To Prior Work:**

There is a good comparison to other primate datasets. A minor remark: Labuguen et al. (2021) contains mostly pictures taken from macaques in outdoor situations in zoos, even though it also contains images from the internet.

**Summary And Contributions:**

The authors propose a dataset of videos and corresponding annotations of chimpanzees recorded between 2015 and 2018 in Leipzig zoo. In total there are two hours of video material, in 163 short videos (average 1000 frames length), both from indoor and outdoor enclosures. The videos are annotated for three tasks: multi animal tracking, pose estimation and spatio-temporal action detection.

The authors provide scores on well-known baseline methods for each task. It is the first longitudinal dataset in a semi-naturalistic and social environment with over 20 chimpanzees and the first primate behavior dataset with ethogram-based action classification.

---

> ### Author Response · Authors · 2023-08-21
>
> We sincerely thank the reviewer for the valuable and constructive feedback. We are happy that the reviewer acknowledges that our dataset fills a gap and has a good explanation of the statistics and that we provide a large number of baselines. In the following, we will address your concerns one by one.
>
> **Q1. generalization into the wild.**
>
> A1. We acknowledge the reviewer's concern that all recordings were obtained from a single zoo and a single group of chimpanzees, which may limit the generalizability of our findings to the wild chimpanzee population. However, we see the longitudinal nature of our dataset as an important contribution to the field. This allows for analyzing behaviors in the context of the long lifespan and complex social environment of these chimpanzees. It is worth mentioning that the group of chimpanzees included in our dataset is the most studied zoo-living chimpanzee group. Extensive research has been conducted on this particular group, resulting in a wealth of knowledge and understanding of their behaviors and social dynamics. By focusing on this well-studied group, we can begin to understand the behaviors and dynamics of this particular group in-depth and contribute further insights. While the semi-natural environment of these apes differs from wild-living populations, we still believe that many of the features of this dataset (e.g., outdoor enclosure with vegetation and challenging light conditions, complex social interactions) will be relevant for studying wild-living chimpanzee populations. We agree that future work should aim to provide longitudinal datasets of multiple groups of the same species.
>
> **Q2. individual identification**
>
> A2. We apologize for any confusion caused. We intended to clarify that our baseline methods primarily focus on multi-object tracking rather than individual identification. While some baseline methods may incorporate re-identification techniques within the tracking process, they do not explicitly support individual identification.
>
> **Q3. Bad performance for pose estimation**
>
> A3. Please refer to **Q4** in the **General Response** for the discussion of poor performance.
>
> **Q4. tracking 10 epochs, pose estimation 210 epoch**
>
> A4. We train all the models following the commonly accepted practice. Training pose estimation models often require more epochs than tracking or action detection tasks in human fields. To address concerns about model performance and overfitting, we have included training or validation curves in the supplementary material (Figure A10). The curves demonstrate that all methods converged, indicating that the models were trained adequately without underfitting or overfitting. Please refer to Section C in the revised supplementary material for more discussion.
>
> **Q5. action methods 2019 or older**
>
> A5. To ensure a fair and comparable baseline, we utilized the MMAction2 framework, which only integrated methods available until 2019. By using a consistent set of methods within a unified training framework, we aimed to establish a reliable basis for performance evaluation and comparison.
>
> **Q6. HOTA**
>
> A6. Thanks and following your suggestion, we've made HOTA the primary metric in the revision.
>
> **Q7. Labuguen et al. (2021)**
>
> A7. Thanks for your correction. We've already corrected Table 1 in our revised manuscript.
>
> **Q8. limited selection of the data**
>
> A8. Please refer to **Q3** in our **General Response**.
>
> **Q9. sample from the complete dataset**
>
> A9. We randomly sample the video clips from the complete dataset.
>
> **Q10. annotation criteria (1) bounding boxes (2) occlusion (3) inter-annotator**
>
> A10. Please first refer to **Q1** in our **General Response**.
>
> (1) The bounding box includes the position (center coordinates, box width, and box height) and the visibility attributes (visible, truncated, occluded).
>
> (2) When an occlusion occurs, we train the annotators to reason about the occluded positions based on the context information and assign the visibility attribute as _occluded_.
>
> (3) To minimize disagreements between annotators, we developed customized annotation guidelines with quantifiable criteria to reduce subjectivity. While some differences may still arise, the labeling company implemented a rigorous process of multiple reviews, rejections, corrections, and further iterations to ensure the smallest possible discrepancies between annotators.
>
> **Q11. videos come from**
>
> A11. Videos were collected and are owned by Prof. Federico Rossano of our group. To date, none of the videos are publicly available. Although the videos were collected years ago, our dataset featuring fine-frained annotations would give public access for the first time. We therefore believe that our dataset would make a unique contribution to the field. The ethics committee of the Wolfgang Köhler Primate Research Center approved the observational data collection for this project.

---

> > ### Comment · Reviewer_TrcR · 2023-08-25
> >
> > Most of my concerns were resolved and I'll increase my rating to 7. Regarding baseline performance: It could be interesting to conduct some hyperparameter tuning (if you haven't done that yet).

---

> > > ### Author Response · Authors · 2023-08-28
> > >
> > > Dear reviewer TrcR,
> > >
> > > Thank you for your reply and for increasing your rating to 7. We appreciate your feedback and are glad to hear that most of your concerns have been resolved.
> > >
> > > Regarding the baseline performance, we have indeed conducted the necessary parameter tuning for some baselines to ensure model convergence. The specific parameters used can be found on our [GitHub repository](https://github.com/ShirleyMaxx/ChimpACT). However, we acknowledge that further hyperparameter tuning could be explored in the next revision.
> > >
> > > Best regards,
> > >
> > > Authors

---

### Official Review · Reviewer_zjvm · 2023-07-21
**Review of ChimpACT - Update**

**Rating:** 7
**Confidence:** 4
**Clarity:** The paper is well written.

**Strengths:**

- This chimp dataset is recorded over a long period of time, and manually labeled for a broad set of tasks (whereas existing primate datasets only focus on a subset of the tasks as reviewed by the authors). The recording conditions also appear to be more natural than indoor recordings.

- The authors perform a large set of experimental evaluations with representative models from the machine learning and computer vision community. The evaluations seem comprehensive.

**Additional Feedback:**

I think this is an interesting dataset, but I'd really like to understand how it might relate given other similar datasets in the video understanding space and clarify some questions about the dataset and experiments. I'm happy discussing more with the authors / changing score if my concerns could be addressed.

**Correctness:**

- The full dataset is not visible to the reviewers (please correct me if I’m wrong).

- The test set seems very small (~2 hours of dataset in total x 10% in test set = results are only on 12 minutes of video). How reliable is the benchmarking?

- There doesn’t seem to be multiple runs / variance computed for the benchmarked methods.

**Documentation:**

- The reviewers don’t have access to the dataset (only 1 sample video is shared I believe?). Please let me know if I've just missed it.

- The experiments were not ran multiple times, and it is difficult to evaluate reproducibility of the benchmark.

**Ethics:**

The paper states that “The ethics committee of the Wolfgang Köhler Primate Research Center approved the observational data collection for this project”.

**Limitations:**

The paper discusses limitations but does not discuss any potential negative societal impacts (authors might want to consider application of behavior models on humans, since some references were made to how this dataset might relate to other human models & datasets (ex: L202))

**Opportunities For Improvement:**

- One concern I have is the size of the dataset with respect to other datasets doing similar tasks (not specifically on chimps). For example, CCR (reviewed by the authors) is larger and covers chimp tracking in the wild; AP-10K [A] has pose estimation over multiple species (not chimp-specific); Calms21 [B] has large sets of annotated mouse behavior with different eval settings. I understand that this is a chimp-specific dataset, but it would be helpful to understand the significance to the broader community of having all these tasks on chimps, since there are many existing datasets for the tasks covered in this dataset ((1) detection, tracking, reID, (2) pose estimation, and (3) action recognition; see more in prior work discussion).

- I'd like more information on the annotation process: what were the expertise of annotators? How were disagreements between annotators resolved? Behavior annotations often produce subjective annotations - how does the authors know their ground truth is correct?

- The videos are recorded over 4 years, but final set is around 2 hours. How were the videos sampled? How were videos sampled into train/val/test (is it random or by time?).

- It's difficult to evaluate the reproducibility of the benchmarking part of the paper since it seems 1 run was performed. It is unclear the significance / variance of methods over multiple runs

[A] Yu, Hang, et al. "Ap-10k: A benchmark for animal pose estimation in the wild." NeurIPS Datasets and Benchmarks (2021).

[B] Sun, Jennifer J., et al. "The multi-agent behavior dataset: Mouse dyadic social interactions." NeurIPS Datasets and Benchmarks (2021).

-------------------

As most of my concerns are addressed by the authors, I have raised my rating. I think the significance of species-specific datasets for the ML community is a general long-term question for benchmarking, but at this time, this dataset has more comprehensive annotations compared to other chimp datasets.

**Relation To Prior Work:**

The prior work in this paper focuses on other non-human-primate datasets but I think the discussion would be improved to include other datasets with similar tasks (ex: there are lots of datasets for pose estimation / tracking / action recognition, which if connections/differences are discussed, could help connect this paper to the wider research community). Some examples of datasets the authors could consider (not comprehensive):

- Datasets from the MOT Challenge (MOT-17 was used in Table 3 I think but would appreciate some further discussions on how MOT relates to ChimpACT tracking/reID since they also have some animal benchmarks like zebrafish): https://motchallenge.net/

- AP-10K: https://github.com/AlexTheBad/AP-10K

- Calms21: https://data.caltech.edu/records/s0vdx-0k302

- Kinetics: https://arxiv.org/pdf/1705.06950.pdf

- APT-36K: https://arxiv.org/pdf/2206.05683.pdf

- Datasets from ActivityNet: http://activity-net.org/

**Summary And Contributions:**

The authors propose a new dataset recorded of a group of chimps in the zoo, focusing on the development of one chimp (Azibo). The dataset consists of ~160k frames, corresponding to ~2 hours of video, recorded from 2015 to 2018. There are 23 different chimp identities in total across the videos. The dataset is annotated manually, for 3 groups of tasks (1) detection, tracking, reID, (2) pose estimation, and (3) action recognition. Finally, for experimental evaluations, the authors evaluate a set of representative existing methods for each type of task on the dataset.

---

> ### Author Response · Authors · 2023-08-21
>
> We sincerely thank the reviewer for the valuable and constructive feedback. We are happy that the reviewer acknowledges that our dataset is recorded over a long period of time and labeled for a broad set of tasks, and we perform a comprehensive evaluation. In the following, we will address your concerns one by one.
>
> **Q1. size of the dataset, all these tasks on chimps**
>
> A1. Please refer to **Q2** in our **General Response**. While there are larger datasets for similar tasks, this chimp-specific dataset is significant because it provides valuable insights into chimpanzee behavior, which is the first dataset that provides both an ethogram and fine-grained chimpanzee behavior labels for a longitudinal dataset. It integrates multiple tasks and allows for a comprehensive understanding of chimpanzee behavior in context. The comprehensive annotations enable the training of models tailored to this chimpanzee population, facilitating focused research and deepening our understanding of their behavior. Note that the focused chimpanzee group in our dataset is one of the most studied primate groups and the most studied zoo-living chimpanzee group.
>
> **Q2. annotation process (1) expertise of annotators (2) disagreements (3) behavior annotations**
>
> A2. Please first refer to **Q1** in our **General Response**.
>
> (1) Our collaborated annotators underwent a rigorous training process to ensure accurate annotations. They received specialized training from the annotation manager, who provided detailed guidelines and instructions based on our customized annotation rules. During the trial annotation phase, the annotators received continuous training and feedback to refine their skills and ensure consistency in their annotations. We have confidence in the proficiency of our annotators due to the thorough training and quality control measures implemented throughout the annotation process.
>
> (2) To minimize disagreements between annotators, we developed customized annotation guidelines with quantifiable criteria to reduce subjectivity. While some differences may still arise, the labeling company implemented a rigorous process of multiple reviews, rejections, corrections, and further iterations to ensure the smallest possible discrepancies between annotators. This iterative annotation process, coupled with thorough quality control measures, helps maintain consistency and minimize variations among different annotators.
>
> (3) To label behaviors, we first supplied example videos showcasing different chimpanzee behaviors, created by our team's experienced primatologists. These videos served as valuable references, enabling annotators to assign behavior labels based on observed actions accurately. Additionally, behavioral primatologists in our team manually reviewed all labeled frames to ensure data reliability.
>
> **Q3. How were the videos sampled? How were videos sampled into train/val/test (is it random or by time?).**
>
> A3. We randomly sample the videos in the large corpus. And the videos are randomly split into train/val/test sets.
>
> **Q4. multiple runs**
>
> A4. We sincerely apologize for only providing the evaluation results with 1 run. During the rebuttal period, we conducted three runs for the experiment and provided the mean and variance of the results in the revision. Although we were only able to complete experiments for tracking due to limited computational resources (see the updated Table 3 in the revised manuscript), we are conducting the pose estimation and action detection experiments in the meantime and will update the results in the next revision. Based on the latest experimental results of Table 3, we observed a relatively small variance, which demonstrates the reproducibility of the benchmarking.
>
> **Q5. any potential negative societal impacts**
>
> A5. Thanks and following your suggestion; we've added more discussion in Section B in the supplementary material. Studying chimpanzees does not raise ethical concerns, as it aligns with ethical standards. However, we emphasize the importance of responsible use of our dataset. We strongly encourage researchers to utilize our dataset solely for research purposes that promote animal welfare and conservation. We firmly discourage any use of the dataset for harmful activities such as hunting or exploitation. It is crucial for researchers to approach the data with a focus on positive societal impacts and to refrain from any potential negative consequences.
>
> **Q6. full dataset**
>
> A6. Please refer to **Q3** in our **General Response**.
>
> **Q7. include other datasets with similar tasks**
>
> A7. Thanks for your suggestion, and we've included more discussion of other datasets in Section 2 in the revision, as you mentioned.

---

> ### Author Response · Authors · 2023-08-28
>
> Dear reviewer zjvm,
>
> Thank you for your reply and for increasing your rating to 7. We appreciate your feedback and are glad to hear that most of your concerns have been resolved.
>
> Best regards,
>
> Authors

---

### Official Review · Reviewer_8sjD · 2023-07-21
**(rebuttal response now at the end) Unique and interesting dataset**

**Rating:** 6
**Confidence:** 4
**Correctness:** See opportunities for improvement.
**Clarity:** Some typos, but overall well written.

**Strengths:**

Datasets such as ChimpACT will be critical for the field to develop sophisticated new methods to study animal behavior in the wild. Such studies will have a range of scientific, and societal, and ecological impacts. These datasets are not trivial to collect and annotate.

**Additional Feedback:**

Please see opportunities for improvement. Overall, I enjoyed the paper and believe it is an important area with unmet needs for new datasets and benchmarks. But I do have some concerns and questions.

--- response to author rebuttal

There are a few concerns that I think the authors still sweep under the rug. If this is accepted, I think it is very important that authors add caveats to some of their claims (e.g., that they have not quantified anything longitudinally, so while it's possible the dataset can enable longitudinal analyses, it's unclear whether the dataset is dense or large enough to actually do so). This context can be added within a discussion of limitations.

**Documentation:**

I have some questions related to the data collection and organization -- see suggestions for improvement.

**Ethics:**

No, all good.

**Limitations:**

Some of the issues mentioned above in opportunities for improvement could be incorporated into the discussion of limitations.

**Opportunities For Improvement:**

I see a few key opportunities for improvement.

- As I mentioned above, these wild animal datasets are not trivial to annotate and can be riddled with artifacts due to the complexity of the environment and occlusions. I feel that the authors currently do not provide adequate summaries of annotation quality, and more details about how annotations were derived across labelers would be helpful. Are multiple 2D pose annotations (from different labelers) averaged to produce a final pose? If so, how consistent are the independently labeled poses, and can variability be used to assess the accuracy of keypoint ground truth? How do the authors know their pose labels are accurate?
- The authors state that every 10th frame in a clip is annotated. Does that mean only ~16k frames are annotated? Or did the group of annotators stagger their labeling to fill in the gaps?
- If I am understanding correctly, each clip is only 1000 frames / 25 fps = 40 seconds long. This doesn’t seem to be very long in the grand scheme of a longitudinal dataset, or even a single action/behavior, which might last minutes.
- The authors highlight the novelty of the longitudinal nature of the dataset, but 1000 clips is a drop in the bucket of 4 years, so it unclear whether the dataset can actually be used for any sort of longitudinal analysis (nor do the authors attempt one).
- Table 1 states that there are “64,289” action labels. If I am understanding correctly, this is the number of frames that are labeled? If so, this number feels quite overinflated and can be misinterpreted at a glance as the number of distinct action *sequences*. With only ~1000 action sequences, to me it is unclear how generalizable methods trained/tested on this dataset will be, especially when considering the real-world use of continuously monitoring and analyzing behavior over a long period of time.
- Related to the last point, how many actions *per day* do these chimpanzees do? Also, are the frames included in this dataset representative of the full behavioral repertoire of these animals? Are there behaviors that are missing from the dataset due to the method used for sampling? Or behaviors that were too difficult to label so they were left out? How did the authors choose their 1000 clips (~2 hours) from the full 405-hour dataset? This is only ~0.5% of the recorded data.
- Can the authors release the entirety of the unlabeled dataset? This would be a huge asset and would address my concerns related to the limited scale of the dataset relative to the 4-year timespan.

**Relation To Prior Work:**

Yes.

**Summary And Contributions:**

The authors present a new dataset, ChimpACT, which contains ~160k annotated video frames collected of a mother-child chimpanzee dyad at the Leipzig zoo over 4 years. Video frames are annotated with bounding boxes, animal IDs, 2D poses, and behaviors, making it one of the most comprehensive ape behavior datasets to date. They also establish benchmarks for tracking, pose estimation, and spatiotemporal action recognition, testing a range of off-the-shelf methods.

---

> ### Author Response · Authors · 2023-08-21
>
> We sincerely thank the reviewer for the valuable and constructive feedback. We are happy that the reviewer acknowledges that our dataset is not trivial to collect and annotate and will be critical to study animal behavior and have a range of scientific impacts. In the following, we address your concerns one by one.
>
> **Q1. annotation quality**
>
> A1. Please refer to **Q1** in our **General Response**.
>
> **Q2. ~16k annotated**
>
> A2. Yes, we manually label around 16K frames. Each frame possesses rich annotations, and labeling 2D keypoints and behaviors is not trivial.
>
> **Q3. 40 seconds**
>
> A3. We would like to clarify that the term "longitudinal" in our dataset does not focus on tracking, pose estimation, or action recognition in a single long-period video. Instead, it pertains to studying the growth and social behaviors of chimpanzees over time, which may last several years. While each clip in our dataset is approximately 40 seconds long, it is important to note that this duration is sufficient for analyzing and identifying specific actions (please see more video samples on our project website). As demonstrated in the provided data examples, meaningful actions can be observed and classified within this timeframe. The longitudinal aspect of our dataset aims to facilitate research on the developmental and social aspects of chimpanzees by combining multiple video segments to study their behaviors and interactions over extended periods.
>
> **Q4. longitudinal**
>
> A4. Our dataset contains videos that were randomly sampled from a span of four years of video footage. While each individual video clip may not exhibit a longitudinal nature on its own, researchers can combine multiple video segments to conduct longitudinal analyses. The dataset's strength lies in its potential to be used collectively, allowing researchers to analyze and track chimpanzee behavior and development across different time points. By considering multiple clips in conjunction, valuable insights into the longitudinal aspects of chimpanzee behavior can be gained.
>
> **Q5. “64,289” action labels**
>
> A5. To clarify the meaning of the "64,289" action labels in Table 1, it is not the count of distinct action sequences. We labeled each frame with an action label for each individual chimpanzee, resulting in the cumulative count of action labels.
>
> We recognize the limitation of the dataset in terms of annotation size. It is indeed a preliminary effort in this domain. However, it is important to note that the dataset provides valuable insights into chimpanzee behavior and serves as an initial exploration in this field. Despite the limited number of action sequences, we encourage researchers to examine the results achieved by methods trained and tested on this dataset. The performance of these methods shows promising similarities to human-centric action recognition tasks, indicating a certain level of generalizability.
>
> **Q6. full behavioral repertoire**
>
> A6. Defining the exact number of actions performed by chimpanzees on a daily basis is challenging and depends on the definition of an action, similar to quantifying human actions. Our ethogram was developed by primatologists and is in line with behaviors commonly captured in wild-living and captive ape groups, focusing on behaviors with social and ethological significance. Our ethogram aims therefore to capture a broad range of social and nonsocial behaviors that are of interest to primatologists. We agree that our ethogram does not capture extremely rare behaviors, like giving birth, or fine-grained behaviors such as facial expressions. The alignment of our annotated behaviors with the proposed ethogram demonstrates the value of the captured data. Our video focal samples were collected across all hours that the chimpanzees had access to their indoor and outdoor enclosures. Apes from this group are moved into smaller subgroups and dormitory structures overnight outside of the hours of our observations. Continuous 24/7 monitoring of idle behaviors is neither necessary nor meaningful and our focal samples cover all hours of day when the ape group is in a shared environment.
>
> Regarding the selection of around 2 hours of video clips from the 405-hour dataset, we employed random sampling to ensure the representation of diverse behaviors and contexts. The chosen clips, while constituting a small percentage (~0.5%) of the recorded data, were able to provide a comprehensive snapshot of the chimpanzees' behavioral repertoire. We invite you to explore and review more video samples on our project website.
>
> **Q7. unlabeled dataset**
>
> A7. Please refer to **Q3** in our **General Response**. We have also added additional data samples with unlabeled video data on the project homepage for your reference. We invite you to explore and review these samples.

---

### Official Review · Reviewer_nT7y · 2023-07-22
**Chimpanzee detection, pose estimation and action labeling dataset with comprehensive set of benchmarks**

**Rating:** 7
**Confidence:** 3
**Correctness:** Yes
**Clarity:** Yes

**Strengths:**

•	The writing is clear and well organized. The manuscript is well-contextualized within the primate ethology field.
•	The comparisons to human SOA algorithms are comprehensive.
•	The dataset contains a rich range of behavioral annotations and a diversity of different annotations


**Additional Feedback:**

* For Longitudinal detection there are likely multiple changes in animal appearance. How is this accounted for and Was detection superior for nearby timepoints?
* Are the number of False Positives and False Negatives really comparable between the human and chimp dataset in Table 3? Should these be normalized to the total number of detections present.
* L278: “These results indicate that chimpanzee pose estimation is a unique and intricate task that cannot be resolved through the mere transfer of human pose estimation algorithms, primarily attributed to two factors: (i) flexible joint articulations and extended range of motion unique to chimpanzees and (ii) the dissimilar physical appearances of their fur in comparison to that of humans.” Is joint flexilibty really the challenge compared to the limited range of training examples, dissimilar appearances and CV factors (illumination conditions etc).
* I would like to see performance of algorithms stratified by the occlusion level of the annotations. Notably I would like to be convinced that algorithms can be performant in the case of unoccluded conditions – if not it is unclear how well-posed the dataset is. Similarly stratifying tracking and pose estimation performance by action type may be helpful, since one expects resting postures to be better tracked.


**Documentation:**

Only a subset of the data is available. There is no discussion of ethical usage or maintenance and there is mention of a datasheet but I do not see one. There should be greater discussion of all of these.

**Ethics:**

See above for a discussion of ethics.

**Limitations:**

From an ethical perspective it could be noted that pose estimation, tracking etc. can be used for surveillance.

**Opportunities For Improvement:**

•	The contextualization with other animal work (flies, mice, etc) is more limited than with primates.
•	The results of algorithms on the dataset are somewhat poor in comparison to human datasets. This is an advantage because it could potentially expose the unique features of animal tracking, but could also result from poor data quality or poor annotations.
•	While the dataset and benchmarks are comprehensive with respect to what is typically done with humans, they don’t address unique challenges of animal identification and the algorithms are not tailored to the potentially unique challenges of the animal identification field.


**Relation To Prior Work:**

* L58: “ChimpACT is the first to offer ethogram annotations for the machine learning and computer vision community”. What about the MABE challenge presented at CVPR (mice, with labels for actions). More generally I would like to see some discussion of other work in the ‘CV for animals’ vein.

**Summary And Contributions:**

This manuscript introduces chimpAct, a dataset composed of two hours of video of monkeys behaving within a Zoo enclosure filmed over the course of several years. The dataset contains videos of multiple different individuals with tracked family relationships, although focusing on a single animal Azibo through a technique called focal sampling. The videos are downsampled from a larger corpus. The data is given high density action labels (ethograms) for both individual and social interactions, as well as bounding boxes, keypoint labels, and visibility estimates for the later two categories. Then present benchmark tasks for tracking and identification, action labeling, and keypoint estimation. They compare the performance of a variety of different state of the art algorithms for each tasks, providing clear benchmarks. They in some cases also provide helpful comparisons with human datasets.  They find state of the art object tracking approaches outperform more classic techniques, locomotion behaviors are better tracked than alternative approaches, and heatmap based pose estimation outperform other approaches including regression based approaches.

---

> ### Author Response · Authors · 2023-08-21
>
> We sincerely thank the reviewer for the valuable and constructive feedback. We are happy that the reviewer acknowledges that our dataset contains rich and diverse annotations, and our paper is well-organized and well-contextualized. In the following, we will address your concerns one by one.
>
> **Q1. other animal work**
>
> A1. While our current focus is primarily on primates, we recognize the importance of incorporating comparative studies across different species to gain broader insights into animal behavior. We've added the discussion of other related animal work in Section 2 in the revised manuscript following your suggestion. We focus on other primate computer vision work because of the unique challenges and features of those datasets. Most work on the other mentioned model species (e.g., flies, rodents) is based on animal behavior in laboratory settings. These settings usually feature simple environments optimized for laboratory research. In contrast, research on primates in wild and semi-natural environments features complex environments in terms of structures, light conditions and social interactions. Further, primates show a broader behavioral repertoire with high flexibility.
>
> **Q2. poor performance, poor annotations.**
>
> A2. Please refer to **Q4** and **Q1** in the **General Response** for the discussion of poor performance, and annotation quality, respectively.
>
> **Q3. unique challenges of animal identification**
>
> A3. We would like to emphasize that our intention was to provide a benchmark that could stimulate interest and attention within the community, ultimately driving advancements in addressing the unique challenges of chimpanzee behavior analysis. And we believe our dataset presents a wealth of opportunities to develop new approaches for solving fundamental tasks in chimpanzee groups, such as animal identification.
>
> **Q4. ethical perspective**
>
> A4. Our goal is to contribute to scientific knowledge and conservation efforts related to chimpanzee behavior analysis. We strongly encourage the community to refrain from using the dataset for unethical research purposes.
>
> **Q5. CV for animals**
>
> A5. Thanks for your suggestion, and we've added more discussion of other "CV for animals" work in Section 2 in the revision.
>
> **Q6. subset of the data**
>
> A6. Please refer to **Q3** in our **General Response**.
>
> **Q7. greater discussion**
>
> A7. In the supplementary Section B, we have included discussions on ethical usage and dataset maintenance. Following your suggestion, we provide a greater discussion as a datasheet in the supplementary Section D.
>
> **Q8. changes in animal appearance**
>
> A8. We did not run a baseline specifically for tracking individuals over extended time spans. Instead, our focus was on tracking within small video segments. However, our dataset provides a valuable resource, and its annotations support conducting research on longitudinal detection and ReID, specifically addressing the challenges associated with tracking animal appearance changes over time.
>
> **Q9. False Positives**
>
> A9. Thanks for your suggestion. We've updated the number of False Positives (FP), False Negatives (FN), and IDs in Table 3 of the revised manuscript to be the normalized ones following your suggestion.
>
> **Q10. joint flexibility**
>
> A10. We speculate that joint flexibility is one of the factors contributing to the difficulty in pose estimation, along with other challenges such as dense fur. However, the fundamental factor may lie in heavy occlusions. Please first refer to **Q2** for a detailed discussion on the challenges of chimpanzee pose estimation. Additionally, we've added Table A6 and Figure A5 in the revised supplementary, which provide evaluations and visualizations of various instances that demonstrate different levels of occlusion in relation to different poses. For resting poses, we often observe significant self-occlusion due to high joint flexibility, resulting in lower PCK scores compared to poses where a larger portion of the body joints is visible, such as climbing. Please refer to Section C.2 in the supplementary material for more qualitative results and discussion.
>
> **Q11. unoccluded conditions**
>
> A11. Thank you for your feedback. Following your suggestion, we have evaluated the performance of the methods specifically for non-occluded poses in the updated supplementary material Table A7. All the methods achieve high accuracy when all the keypoints are visible. This demonstrates their effectiveness in accurately estimating poses and also highlights the high quality of our data annotations. Additionally, we evaluated the performance with regard to different behavior categories in Table A5 in the supplementary material as you suggested, and we observe that different action types exhibit variations in pose accuracy. Please refer to Section C.2 in the supplementary material for more qualitative results and discussion.

---

> > ### Comment · Reviewer_nT7y · 2023-08-27
> >
> > I appreciate the very detailed reply to myself and the other reviewers. It is nice to see that most reviewers appreciated the contribution and the difficulty of the task. I'll keep the score the same. A few notes:
> >
> > > Q3. unique challenges of animal identification
> > - It seems like this could be a point to discuss occlusion rich environments with limited pretraining.

---

> > > ### Author Response · Authors · 2023-08-28
> > >
> > > Dear reviewer nT7y,
> > >
> > > Thanks for your reply and the valuable suggestions. We will add a discussion in the revision to encourage exploring how limited pretraining can effectively generalize to occluded environments.
> > >
> > > Best regards,
> > >
> > > Authors

---

### Official Review · Reviewer_H6yA · 2023-07-22
**I have read the rebuttal of the authors**

**Rating:** 6
**Confidence:** 4
**Correctness:** the claims made in the submission are…
**Clarity:** The writing and the figures can be im…

**Strengths:**

1. The dataset on animals is very hard to obtain.
2. The paper is well organized.
3. The proposed dataset is interesting.

**Additional Feedback:**

I have read the rebuttal of the authors and the comments of other reviewers. Although the rebuttal does not actually fully addresses my concerns, animal dataset is valuable. I choose to raise my score.

**Documentation:**

Yes, sufficient detail are provided.

**Limitations:**

1. The dataset is 2D. I understand that the animal dataset is very hard to obtain, however, 2D limits its usefulness.
2. The dataset size is not large.
3. The writing and the figures can be improved.

**Opportunities For Improvement:**

1. The dataset is 2D. I understand that the animal dataset is very hard to obtain, however, 2D limits its usefulness.
2. The dataset size is not large.
3. The writing and the figures can be improved.

**Relation To Prior Work:**

Not that clear.

**Summary And Contributions:**

The paper presents a dataset for understanding chimpanzee behaviors. The proposed dataset is interesting. The paper is well organized.

---

> ### Author Response · Authors · 2023-08-21
>
> We sincerely thank the reviewer for the valuable and constructive feedback. We are happy that the reviewer acknowledges that our dataset is interesting, very hard to obtain and the paper is well organized. In the following, we will address your concerns one by one.
>
> **Q1. The dataset is 2D. I understand that the animal dataset is very hard to obtain, however, 2D limits its usefulness.**
>
> A1. We appreciate the reviewer's concern about the dataset being 2D. Nonetheless, for our investigation of chimpanzee behaviors, the application of 2D bounding box, pose, and action labels proves to be quite adequate. Obtaining a comprehensive 3D dataset for chimpanzees is challenging, and our focus is on analyzing behavioral patterns and interactions rather than precise spatial relationships. The 2D data, combined with advanced computer vision techniques, can still provide meaningful insights into chimpanzee behavior. We believe that our research objectives are justified and will contribute to the understanding of chimpanzee behaviors.
>
> **Q2. The dataset size is not large.**
>
> A2. Please see **Q2** in our **General Response**.
>
> **Q3. The writing and the figures can be improved.**
>
> A3. We are committed to enhancing the clarity and quality of our paper and would greatly appreciate it if more specific details or suggestions for improvement can be provided. Without specific guidance, it becomes challenging to address the concerns effectively.

---

> ### Author Response · Authors · 2023-08-28
>
> Dear reviewer H6yA,
>
> Thanks for your reply and raising the score. Should there be any concerns left, we welcome your comments and are open for discussion at any time.
>
>
> Best regards,
>
> Authors

---

### Author Response · Authors · 2023-08-21
**General Response from Authors**

# General Response

We sincerely thank all reviewers for dedicating your time and effort to the meticulous review of our manuscript. We greatly appreciate that all reviewers acknowledge that (i) our dataset provides **rich annotations** (Reviewer **nT7y**), is **interesting** (Reviewers **H6yA**, **8sjD**, **zjvm**, and **TrcR**), and offer **comprehensive** evaluations (Reviewers **nT7y**, **zjvm**, and **TrcR**); and (ii) our paper is **well-organized** (Reviewers **H6yA** and **nT7y**) or **well-written** (Reviewers **8sjD**, **zjvm**, and **TrcR**). In this general response, we will first answer the common concerns raised by the reviewers. We answer each reviewer's individual questions in separate responses.

**Q1. Annotation process and quality.**

A1. We apologize for the lack of detail in our previous manuscript. Due to the space limit, please refer to Section A.2 of the supplementary material for a more detailed overview of the annotation process and quality control. With the standard and strict measures and the involvement of primatologists, the overall quality and reliability of the annotations are guaranteed.

**Q2. The size of the dataset.**

A2. We would like to emphasize that our dataset is currently the largest available for chimpanzee behavior analysis and pose estimation. Obtaining accurate labels for actions and poses in the context of chimpanzees is more challenging than bounding boxes, making our dataset a significant contribution to the field.

**Q3. The full dataset is not visible.**

A3. We apologize for being unable to provide the complete dataset at this time. Given the efforts our team has spent on collecting, labeling, and training, we hope this is understandable. We assure you that upon acceptance of our work, we will make the full annotated dataset publicly available. In the meantime, we have added additional data samples on the project homepage for your reference. We invite you to explore and review these samples.

**Q4. Poor performance.**

A4. We have analyzed the reasons for the inferior performance in the experiment section (Section 4). Here, we provide further quantitative analysis and have included these analysis results in the revised supplementary materials (especially Section C.2).

The underperformance of algorithms on our dataset may be attributed to two main factors. First, the smaller size of our ChimpACT dataset, with only around 60K annotations, compared to larger human datasets like COCO (250K pose annotations) [a], may impact the model's performance. Second, chimpanzee detection and pose estimation pose unique challenges due to their distinct living environment and behaviors. Chimpanzees inhabit complex habitats with frequent occlusions caused by tree branches and ropes (see more video samples on our project website). Their social nature leads to close proximity and occlusions between individuals, making detection more difficult. Moreover, chimpanzees have greater joint mobility and a wider range of motion, adding complexity to pose estimation. The presence of self-occlusion, combined with their similar fur coloration, further hinders accurate pose estimation. We evaluated pose estimation accuracy in Table A7 in the updated supplementary material, where all methods achieved high accuracy when keypoints were fully visible, highlighting the effectiveness of the pose algorithms. The excellent performance of the algorithms in the absence of occlusions highlights the high quality of our data annotations. However, the current methods' poor performance on our dataset further confirms the challenging and unique nature of animal perception. Please refer to Section C.2 in the supplementary material for more qualitative results and discussion.

**Revision notes**

To facilitate the reviewer's quick identification of our revisions, we have compiled a brief list of the modifications made:

- **Main Text - Related Work:** Add more discussion on related work regarding animals and humans in similar tasks.

- **Main Text - Experiments:** Include multiple run results on the tracking task and discussion in Section 4.1 and Table 3.

- **Supplementary Material - Section A.2:** Provide a detailed discussion on the annotation process and quality.

- **Supplementary Material - Section B:** Add a discussion on intended uses and potential negative impacts.

- **Supplementary Material - Section C:** Include training curve graph A10 and its discussion. In Section C.2, add experimental results for pose estimation considering factors like non-occlusion. Also, add a discussion on cases of bad performance. Additionally, we include Table A5, A6, A7, and Figure A5.


[a] Lin, Tsung-Yi, et al. "Microsoft coco: Common objects in context." ECCV. 2014.

---

### Decision · Program_Chairs · 2023-09-22

**Decision:**

Accept (Poster)

**Comment:**

All reviewers rate the submission above the acceptance threshold and they acknowledge the importance of the dataset for the study of chimpanzee social behavior in a semi-naturalistic environment. The challenges posed by the collection and annotation of such a benchmark are well addressed in this work. The authors are advised to make good use of the thoughtful recommendations provided by reviewers, especially with regard to discussion of: 1) occlusion handling, 2) suitability of the data for longitudinal analysis, 3) similarity to existing animal behavior datasets, 4) model selection and hyperparameter tuning.